# Engraftment of allogeneic iPS cell-derived cartilage organoid in a primate model of articular cartilage defect

Kengo Abe[1,2,3], Akihiro Yamashita[1,3], Miho Morioka[1], Nanao Horike[1,3], Yoshiaki Takei[3,4], Saeko Koyamatsu[1], Keisuke Okita [5], Shuichi Matsuda [2] & Noriyuki Tsumaki [1,3,6] ✉

Induced pluripotent stem cells (iPSCs) are a promising resource for allogeneic cartilage transplantation to treat articular cartilage defects that do not heal spontaneously and often progress to debilitating conditions, such as osteoarthritis. However, to the best of our knowledge, allogeneic cartilage transplantation into primate models has never been assessed. Here, we show that allogeneic iPSC-derived cartilage organoids survive and integrate as well as are remodeled as articular cartilage in a primate model of chondral defects in the knee joints. Histological analysis revealed that allogeneic iPSC-derived cartilage organoids in chondral defects elicited no immune reaction and directly contributed to tissue repair for at least four months. iPSC-derived cartilage organoids integrated with the host native articular cartilage and prevented degeneration of the surrounding cartilage. Single-cell RNA-sequence analysis indicated that iPSC-derived cartilage organoids differentiated after transplantation, acquiring expression of PRG4 crucial for joint lubrication. Pathway analysis suggested the involvement of SIK3 inactivation. Our study outcomes suggest that allogeneic transplantation of iPSC-derived cartilage organoids may be clinically applicable for the treatment of patients with chondral defects of the articular cartilage; however further assessment of functional recovery long term after load bearing injuries is required.

Articular cartilage covers the ends of bones and provides lubrication, which is vital for smooth joint movement and shock absorption. Articular cartilage is avascular and consists of chondrocytes embedded in the extracellular matrix (ECM), which enables the mechanical functions necessary for joint motion and shock absorption. Cartilage ECM consists of collagen fibrils composed of type II, IX, and XI collagen molecules and proteoglycans composed of aggrecan, link protein, and glycosaminoglycans.

As articular cartilage has a limited capacity for repair, and so far, no drugs are available for cartilage repair, focal damage or erosion of articular cartilage frequently leads to debilitating conditions, such as osteoarthritis. Although cell-based therapies have been proposed, only a limited number of autologous chondrocytes are generated since expansion culture steers the chondrocyte character toward that of fibroblastic cells[1], as evidenced in autologous chondrocyte implantation (ACI), wherein more than 90% of the repaired tissue is

[1]Department of Tissue Biochemistry, Graduate School of Medicine and Frontier Biosciences, Osaka University, Osaka, Japan. [2]Department of Orthopaedic Surgery, Graduate School of Medicine, Kyoto University, Kyoto, Japan. [3]Department of Clinical Application, Center for iPS Cell Research and Application, Kyoto University, Kyoto, Japan. [4]Regenerative Medicine Technology Department, Healthcare R&D Center, Asahi Kasei Corporation, Kyoto, Japan. [5]Department of Life Science Frontiers, Center for iPS Cell Research and Application, Kyoto University, Kyoto, Japan. [6]Premium Research Institute for Human Metaverse Medicine (WPI-PRIMe), Osaka University, Osaka, Japan. ✉e-mail: ntsumaki@tsu.med.osaka-u.ac.jp

fibrocartilaginous[2]. Allogeneic cartilage has been transplanted clinically without matching human leukocyte antigen (HLA) types and without the use of immunosuppressive drugs[3–5]. However, whether the transplanted allogeneic cartilage causes an immune reaction remains controversial. Some reports suggest low immunogenicity of chondrocytes[6,7], whereas others show that chondrocytes are antigenic and elicit varying degrees of immune reactions[8,9]. To the best of our knowledge, the regenerative mechanisms following allogeneic cartilage transplantation have not yet been reported. Whether transplanted cartilage achieves engraftment (survives and constitutes repaired tissue directly) or only transiently remains and secretes growth factors to stimulate recipient progenitor cells has not been analyzed.

Induced pluripotent stem (iPS) cells are a promising source for the regenerative treatment of articular cartilage damage[10,11]. Cartilage consisting of chondrocytes and ECM has been successfully created from iPS cells by differentiating them into chondrocytes, which are subsequently transferred into a three-dimensional culture to make iPS cell-derived chondrocytes produce and accumulate ECM around themselves to form cartilaginous tissue particles[12,13]. Owing to the self-renewal activity of iPS cells, allogeneic iPS cell-derived cartilage organoids can theoretically be produced inexhaustibly and transplanted into an unlimited number of patients, solving the issues associated with allogenic cartilage, such as scarcity of donors, risk of disease transmission, and variations in cartilage qualities between donors.

In this study, we analyzed the allogenic transplantation of major histocompatibility complex (MHC)-mismatched iPS cell-derived cartilage organoids in a primate animal model without the use of immunosuppressive drugs. We differentiated *cynomolgus monkey* iPS cells (cyiPSCs) into chondrocytes to create cyiPSC-derived cartilage organoids (cyiPS-Cart). We then transplanted cyiPS-Cart into chondral defects on the knee joint surface of *cynomolgus monkeys* in an allogeneic manner. Single-cell RNA-sequencing (scRNA-seq) and molecular analysis of the cyiPS-Cart graft revealed molecular pathways involved in cell differentiation that remodeled the cyiPS-Cart toward articular cartilage after transplantation.

## Results

### Preparation of cyiPS cell-derived cartilage organoid (cyiPS-Cart)
1466A1 cyiPSCs expressing enhanced green fluorescent protein (EGFP) under a constitutive promoter were used. Cartilage was created from cyiPSCs[14] and human iPSCs[12,13] using modified protocols. Briefly, chondrocytes were induced from cyiPSCs in a chondrogenic medium for two weeks and transferred to a three-dimensional culture, where they produced and accumulated ECM to form cartilaginous particles (Supplementary Fig. 1a). The cyiPSC-derived cartilage organoid (cyiPS-Cart) particles were 1–3 mm in diameter (Supplementary Fig. 1b). Histological analysis showed that the particles consisted of cells, and the ECM was stained positively with safranin O. Immunohistochemical analysis revealed that the ECM contained type II collagen (Supplementary Fig. 1c). Type I collagen was not detectable except at the periphery of a particle.

### Allogeneic transplantation of cyiPS-Cart in primate chondral defect model
Cartilage defects can be classified into two categories based on their depth: chondral defects extending down to but not through the subchondral bone, and osteochondral defects extending down through the subchondral bone (Fig. 1a). Chondral defects are the most common in patients with articular cartilage damage or erosion, including during the early stages of osteoarthritis.

We created chondral defects in the femoral trochlear ridge of the right knee joints of 12 *cynomolgus monkeys* and transplanted cyiPS-Cart (transplantation group) in six monkeys or nothing (empty group) in the remaining monkeys (Fig. 1b). MHC typing revealed a mismatch between cyiPSCs and recipient monkeys (Supplementary Table 1).

Computed tomography (CT) imaging analysis of the knee joints indicated that bone structures were normal immediately after surgery and at 4 and 12 weeks after surgery (Supplementary Fig. 2), suggesting that the defects were chondral and did not extend through the subchondral bone throughout the experiment.

Three monkeys from each group were sacrificed 4 and 17 weeks after surgery ($n = 3$). The gross appearance of the joint surface indicated that chondral defects in the empty group were filled with brown tissue at 4 and 17 weeks (Fig. 1c). On the other hand, chondral defects in the transplantation group were filled with transparent tissue at 4 weeks (Fig. 1c) which later turned white as articular cartilage, making it difficult to distinguish the area where cyiPS-Cart were transplanted from the surrounding articular cartilage area at 17 weeks after transplantation (Fig. 1c). In each monkey, one transplant site was harvested and subjected to histological analysis and other two sites were combined and used for scRNA-seq analysis.

### Allogeneic transplantation of cyiPS-Cart did not elicit an immune reaction in primate chondral defects
We recently reported that allogeneic cyiPS-Cart elicited an immune reaction when transplanted into osteochondral defects[14]. However, no scientific evidence exists on whether allogeneic cartilage elicits an immune reaction when implanted in chondral defects. To answer this question, we analyzed histological sections of the transplanted sites. For control, we created osteochondral defects in additional three monkeys, transplanted cyiPS-Cart, and sacrificed them 4 weeks later.

Four weeks after allogeneic transplantation into osteochondral defects, histological analysis revealed that many cells, including CD3+ T lymphocytes, accumulated around the cyiPS-Cart in the bone marrow (Fig. 2a, b), which is consistent with a previous report[14]. In contrast, there was no cell accumulation around chondral defects transplanted with allogeneic cyiPS-Cart (Fig. 2a, b).

### cyiPS-Cart survived and directly contributed to hyaline cartilage-rich repaired tissue in chondral defects
The quality of repaired tissues in the chondral defects was assessed by staining histological sections with safranin O, which stains the cartilaginous proteoglycans. Chondral defects in the empty group were partially filled at 4 weeks and substantially filled at 17 weeks after surgery, with tissues that were not cartilaginous but fibrous, as indicated by negative safranin O staining (Fig. 3a and Supplementary Figs. 3, 4a). In contrast, chondral defects in the transplantation group were filled with cartilaginous tissue both at 4 and 17 weeks after surgery, as indicated by positive safranin O staining (Fig. 3a and Supplementary Figs. 3, 4a).

Repaired tissues formed in the chondral defects in the empty groups at both 4- and 17 -weeks after transplantation showed positive picrosirius red staining under polarized microscopy (Fig. 3a), suggesting that the repaired tissues were fibrous. On the other hand, marginal staining was observed in the repaired tissues that filled chondral defects and the articular cartilage in the transplantation groups at both 4- and 17- weeks after transplantation (Fig. 3a), suggesting that they were hyaline cartilage.

Scoring cartilage repair (Supplementary Table 2) revealed better cartilage regeneration in the transplantation group than in the empty group (Fig. 3b). The score improved at 17 weeks from 4 weeks after transplantation (Fig. 3b).

The articular cartilage adjacent to the chondral defect in the empty group lost safranin O staining at 17 weeks after surgery (Fig. 3a, arrows and Supplementary Fig. 4b). The remaining cartilage locates between the bottom of the defect and bone also lost safranin O staining at 4 and 17 weeks after surgery in the empty group (Fig. 3c, area below the dotted lines). These results indicate progressive degeneration of the articular cartilage around the defects. In contrast, the articular cartilage surrounding the chondral defect maintained

proteoglycan in the transplantation group at 17 weeks after surgery, suggesting preservation of articular cartilage around the defects (Fig. 3a, arrowheads; Fig. 3c, area below the dotted lines; and Supplementary Fig. 4b).

Immunostaining with anti-GFP antibody revealed that almost all cells in the repaired tissues that filled the chondral defects in the transplantation group expressed GFP (Fig. 4) at both 4 and 17 weeks after transplantation, indicating that transplanted cyiPS-Cart survived and directly contributed to the entire repaired tissue for at least 4 months. Additional immunostaining with antibodies that recognize type I collagen (COL1), a marker for fibrous tissue, and type II collagen (COL2), a marker for cartilage, confirmed that the surviving cyiPS-Cart filling chondral defects contained hyaline cartilage. In contrast, the tissues filling empty chondral defects were fibrous (Fig. 4).

The integration of repaired tissue and surrounding native cartilage is difficult, especially in the case of chondral defects, because chondral defects do not bleed[1]. In our study, cyiPS-Cart did not achieve integration at 4 weeks, but integration was observed 17 weeks after transplantation. Although immature tissue was still bridging the cyiPS-

Cart and surrounding native cartilage, it was mainly composed of type II collagen (Fig. 4, arrows).

No accumulation of immune cells, including CD3+ T lymphocytes, was observed in the transplantation group 17 weeks after allogeneic transplantation into osteochondral defects (Figs. 3a, 4b), indicating that allogeneic transplantation of cyiPS-Cart into chondral defects did not elicit immune reactions in the primate model for at least 4 months.

**Post-transplant cyiPS-Cart remains cartilaginous while tissues formed in chondral defects in the empty group were fibrous**

To investigate the fate of cyiPS-Carts after transplantation, we performed scRNA-seq analysis. Single cells were prepared from undifferentiated cyiPSCs (cyiPSC), cyiPS-Cart (pre-transplant cyiPS-Cart), intact articular cartilage (cyAC), fibrous tissue formed in chondral defects in the empty group (cyFT), and cyiPS-Cart in chondral defects in the transplantation group (post-transplant cyiPS-Cart) 17 weeks after surgery (Fig. 5a).

As for chondrogenic differentiation of cyiPSCs toward pre-transplant cyiPS-Cart, the scRNA-seq analysis revealed that cyiPSCs expressed pluripotency markers, whereas cells in pre-transplant cyiPS-

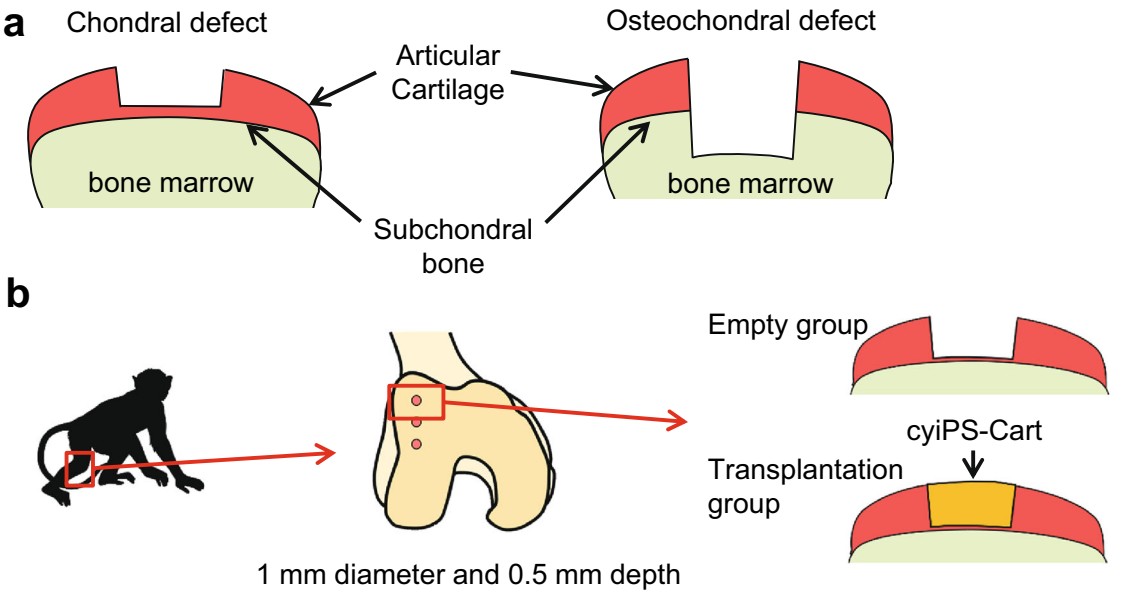

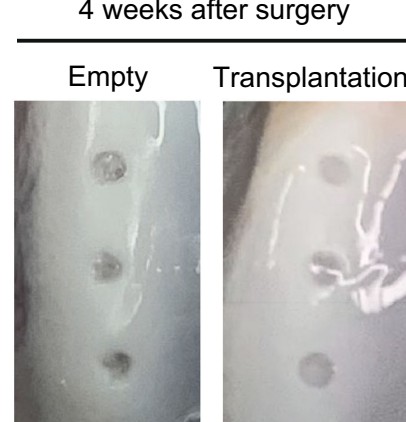

**Fig. 1 | Transplantation of cyiPS-Cart in the knee joints in a primate model.** **a** Two categories of articular cartilage defect. Left: chondral defects extending down to but not through the subchondral bone. Right: osteochondral defects extending down through the subchondral bone. **b** Primate model for cyiPS-Cart transplantation. Chondral defects were created in the femoral trochlear ridge of the right knee joints in *cynomolgus monkeys*. CyiPS-Cart (transplantation group) or nothing (empty group) were transplanted into the defects. The monkey image is taken from [https://www.flaticon.com/free-icon/monkey_47138] following the Flaticon license guidelines. **c** Gross appearance of the joint surface 4 weeks (left) and 17 weeks (right) after surgery. Data were representative of three monkeys.

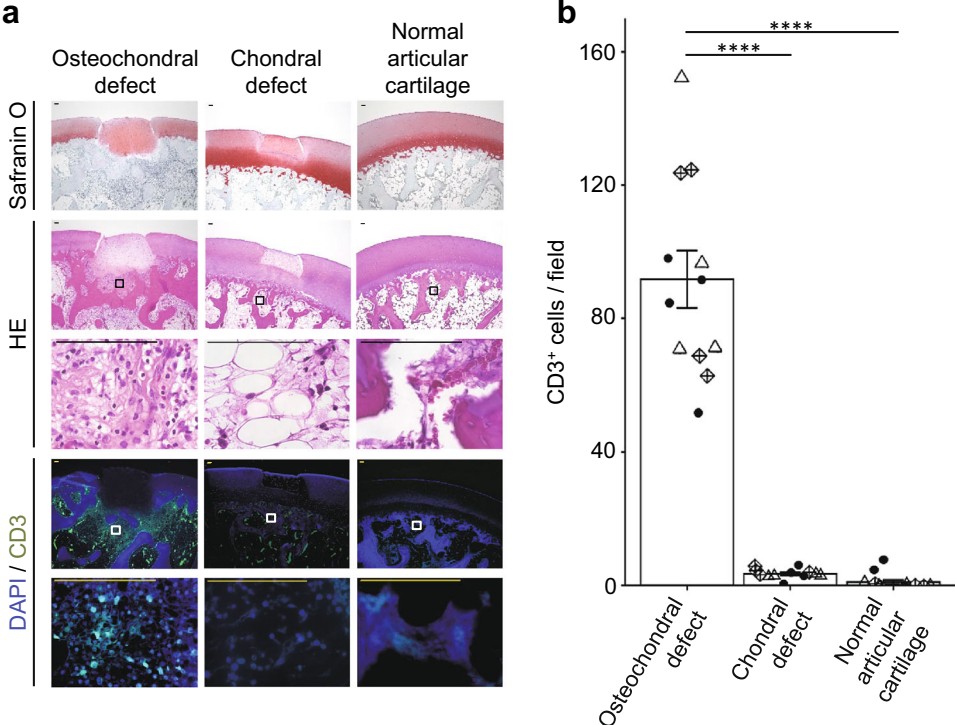

**Fig. 2 | Analysis of immune reactions following allogeneic transplantation of cyiPS-Cart into chondral defects in the knee joints 4 weeks after transplantation. a** Semi-serial histological sections were stained with safranin O or HE, or immunostained for CD3. Scale bars, 100 μm. **b** The number of CD3+ cells per microscopic field was determined. Four fields were used for each monkey. Three monkeys were used in each group. Each mark indicates one field, and different shapes of marks indicate different monkeys. Error bars denote mean ± SE. ****P < 0.0001 by one-way ANOVA with post hoc Tukey HSD test (*n* = 12 fields). Source data are provided as a Source Data file.

Cart did not express pluripotent markers but expressed chondrocyte markers (Fig. 5b). Some of the pre-transplant cyiPS-Cart cells expressed *COL1A1* (Fig. 5b) and likely resided in the periphery of the pre-transplant cyiPS-Cart, as indicated by COL1 immunostaining (Supplementary Fig. 1c). All cyiPSCs and pre-transplanted cyiPS-Cart cells expressed EGFP. The uniform manifold approximation and projection (UMAP) plot indicated that cyiPSCs and pre-transplanted cyiPS-Cart cells were plotted as separate clusters (Fig. 5c). Featureplot analysis revealed that the pre-transplant cyiPS-Cart cell cluster expressed chondrocyte markers but not pluripotent markers (Fig. 5d), indicating chondrogenic differentiation of cyiPSCs into pre-transplant cyiPS-Cart cells.

For samples harvested at 17 weeks after surgery, VlnPlot function (Seurat) revealed positive GFP expression in almost all cells in the post-transplant cyiPS-Cart (Fig. 6a), confirming that cyiPS-Cart survived and directly contributed to the repaired tissue. Almost all cells in post-transplant cyiPS-Cart expressed *COL2A1* and not *COL1A1*, whereas the majority of cells in cyFT expressed *COL1A1* but not *COL2A1* (Fig. 6a), confirming that post-transplant cyiPS-Cart remained hyaline cartilaginous while the cyFT was fibrous.

Then, we analyzed cell subpopulations in cyAC, cyFT, pre-transplant cyiPS-Cart, and post-transplant cyiPS-Cart samples. We reduced cell numbers to 320 in each sample using a subset function and integrated samples into a single object[15]. We also reduced dimensions, clustered the cells with a parameter resolution of 0.2, and projected them onto a UMAP plot (Fig. 6b). Cell clustering analysis revealed that cyAC, pre-transplant cyiPS-Cart, and post-transplant cyiPS-Cart had similar transcriptional profiles, whereas FT contained cell clusters with distinct profiles. cyAC, pre-transplant cyiPS-Cart, and post-transplant cyiPS-Cart were composed of cluster # 0, whereas FT was composed of clusters # 1 and # 2 (Fig. 6c, d). Cells in clusters #0 and #2 exhibited high expression of *COL2A1* whereas those in cluster

#1 highly expressed *COL1A1* (Fig. 6e). Differentially expressed genes (DEGs) were identified (Fig. 6f). Canonical pathway analysis based on the DEGs indicated that cluster #1 was enriched for the fibrosis pathway and that cluster # 2 was enriched for the osteoarthritis pathway (Fig. 6g). These results suggest that clusters #1 and #2 consisted of pathological cells and contain few cyAC, pre-transplant cyiPS-Cart, and post-transplant cyiPS-Cart cells (Fig. 6d). We further compared these cell clusters with those previously identified in human osteoarthritis samples[16]. The expression of marker genes for osteoarthritis[16] in our clusters suggested that cluster #1 corresponds to preHTC and FC (high expression of *TGFBI and COL1A1*) and that cluster #2 corresponds to EC and proC (high expression of *TF* and *P3H2*) (Supplementary Fig. 5a, b).

To analyze the resemblance and difference between cyAC and post-transplant cyiPS-Cart cells, we selected these cells and performed clustering analysis again. Cells were divided into four clusters (Supplementary Fig. 6a, b). Clusters #0, 1 and 2 consisted of both cyAC and post-transplant cyiPS-Cart cells, whereas post-transplant cyiPS-Cart cells were excluded from cluster #3 (Supplementary Fig. 6a, c, d). DEGs and canonical pathway analysis indicated that cluster #3 was enriched for integrin signaling (Supplementary Fig. 6e, f). Trajectory inference and RNA velocity analysis suggested that cluster #3 was located at the start of the trajectory to #0 (Supplementary Fig. 6g), whereas cluster #2 was located at the end. These results suggest that post-transplant cyiPS-Cart are similar to cyAC except for cells from cluster #3 that are related to integrin signaling and locate at the start of the trajectory.

**cyiPS-Cart became similar to articular cartilage after transplantation, with cells differentiated to express *PRG4*, achieving engraftment**

Next, we examined how the nature of cyiPS-Cart was altered after transplantation into a chondral defect. Among the differentially

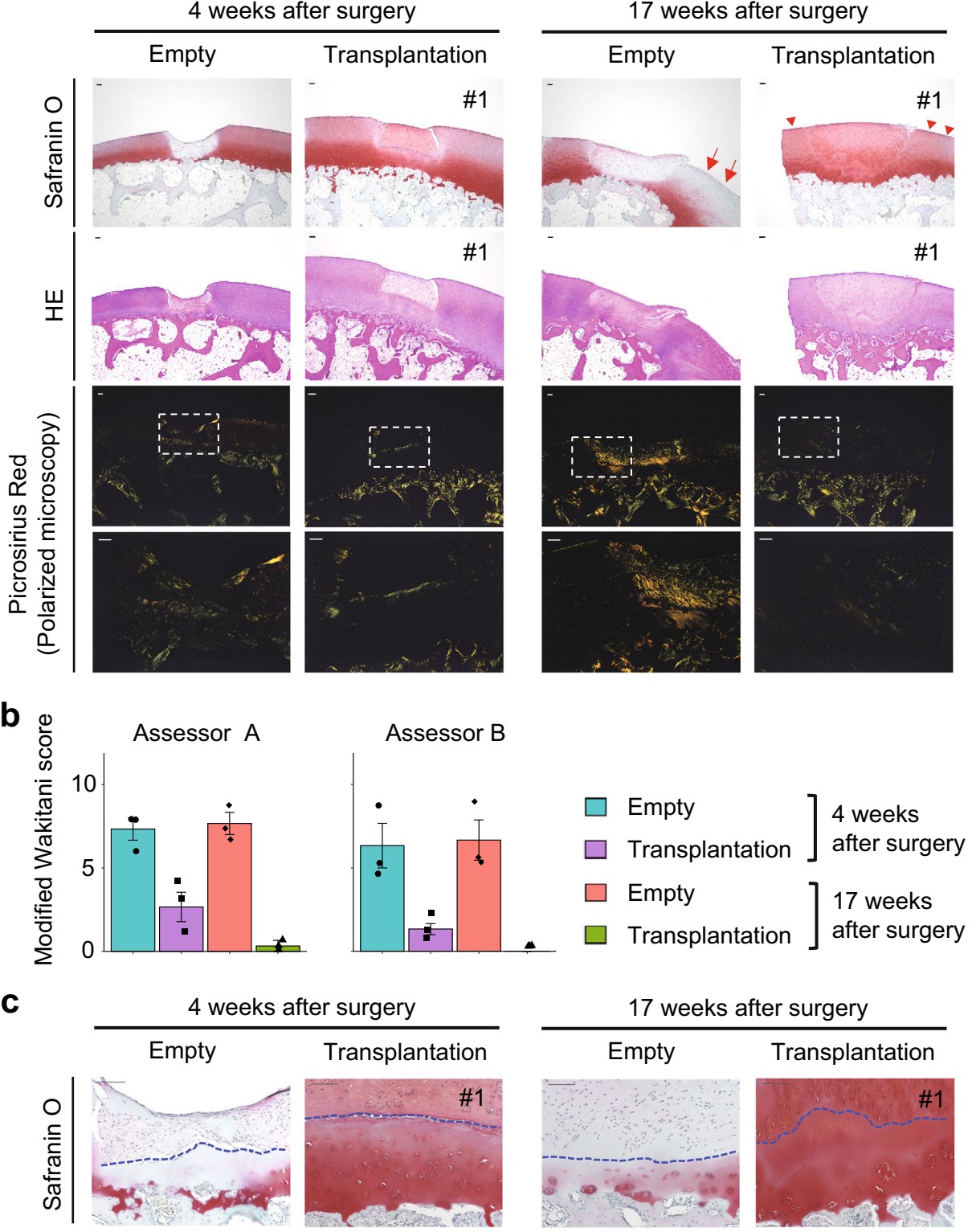

**Fig. 3 | Qualities of repaired tissue in chondral defects. a** Samples were harvested 4 or 17 weeks after transplantation. Semi-serial sections were stained with Safranin O, HE, or picrosirius red. Sections stained with picrosirius red were observed under a polarized microscope. A magnification of the boxed regions that cover repaired tissue and native articular cartilage in the third row is indicated in the bottom row. Scale bars, 100 μm. **b** Sections stained with Safranin O were subjected to a modified Wakitani histological scoring system ($n = 3$ monkeys in each group) and evaluated by two independent assessors in a blinded manner. Error bars denote mean ± SE. Source data are provided as a Source Data file. **c** Magnifications of remaining cartilage located between the bottom of the defect and bone in (**a**). Dotted lines indicate the bottom of defects. Safranin O staining. Scale bars, 100 μm.

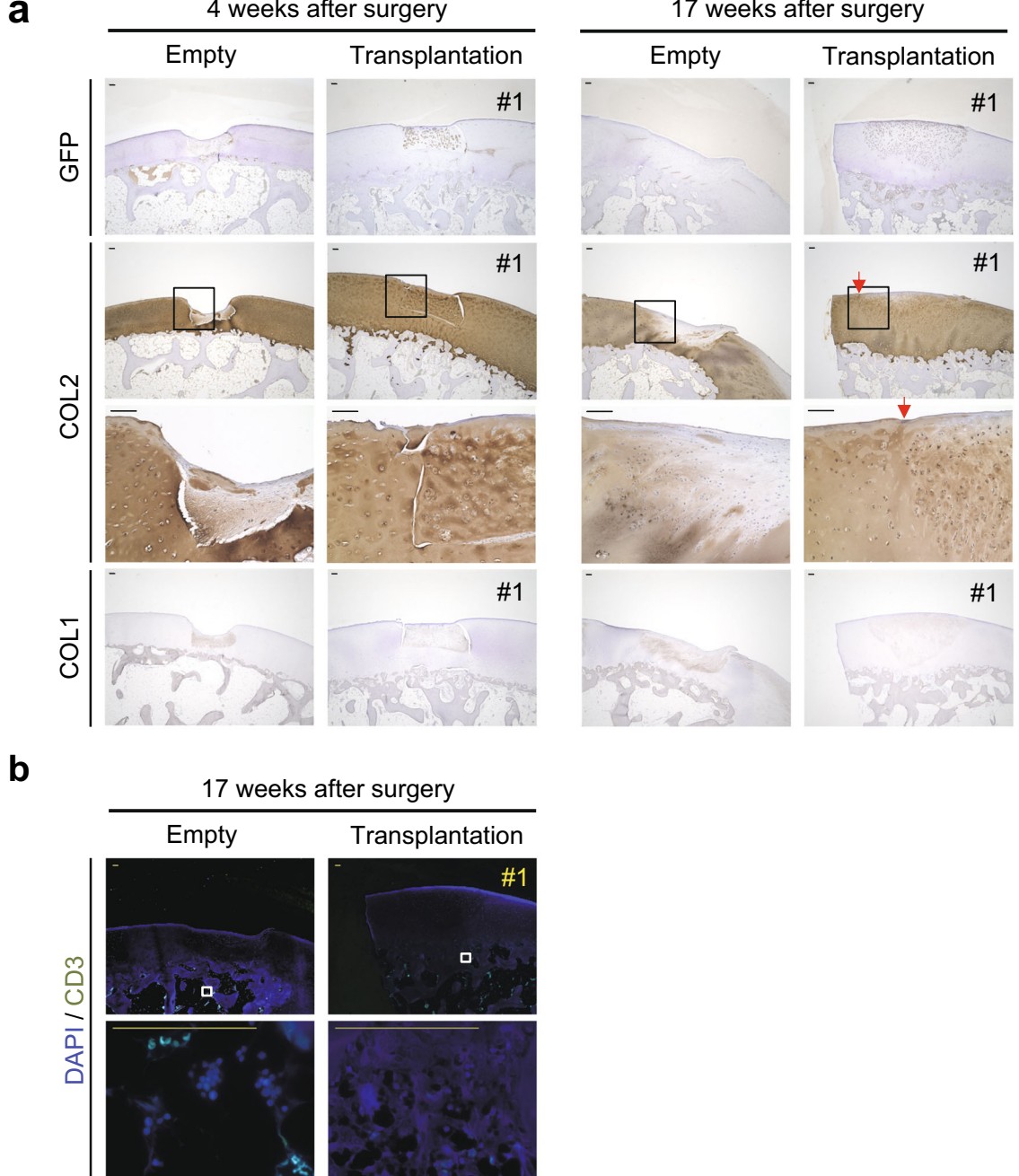

**Fig. 4 | Immunohistochemical staining of repaired chondral defects. a** Samples were harvested 4 or 17 weeks after transplantation. Semi-serial sections were immunostained for GFP, type II collagen (COL2), and type I collagen (COL1). The boxed regions in the second row are magnified in the third row. Data were representative of three monkeys. **b** Semi-serial histological sections of samples at 17 weeks after transplantation were immunostained for CD3. Data are representative of three monkeys. Scale bars, 100 μm.

expressed genes between pre-and post-transplant cyiPS-Cart, the expression of gene encoding proteoglycan 4 (*PRG4*) was significantly increased (adjusted *p* value = 1.55 × 10⁻²⁸) (Fig. 7a). PRG4 is expressed in the superficial zone of articular cartilage and crucial for the lubrication of the joint surface[17–19]. The VlnPlot function (Seurat) revealed that few cells expressed *PRG4* in pre-transplant cyiPS-Cart, while a substantial number of post-transplant cyiPS-Cart cells showed high-level *PRG4* expression compared to cyAC cells (Fig. 7b). FeaturePlot function (Seurat) confirmed that few cells in the pre-transplant cyiPS-Cart expressed *PRG4* whereas many cells in the post-transplant cyiPS-Cart expressed *PRG4* (Fig. 7c). Immunohistochemical analysis revealed that *PRG4* expression was hardly detected in pre-transplant cyiPS-Cart,

consistent with the low expression of *PRG4* mRNA observed in scRNA-seq analysis (Fig. 7d). In contrast, post-transplant cyiPS-Cart expressed *PRG4*, and its expression was localized to the superficial zone of post-transplant cyiPS-Cart. This expression pattern was consistent with that in cyAC (Fig. 7d).

To gain further insights into the mechanism by which cyiPS-Cart acquired *PRG4* expression, we analyzed scRNA-seq data. We combined pre-and post-transplant cyiPS-Cart cells and separated them into two groups: cells whose *PRG4* expression was more than or equal to 2.4, and those whose *PRG4* expression was less than 2.4. We then identified differentially expressed genes (DEGs) between the two groups using the FindMarkers function (Seurat) (Supplementary Data 1) and subjected

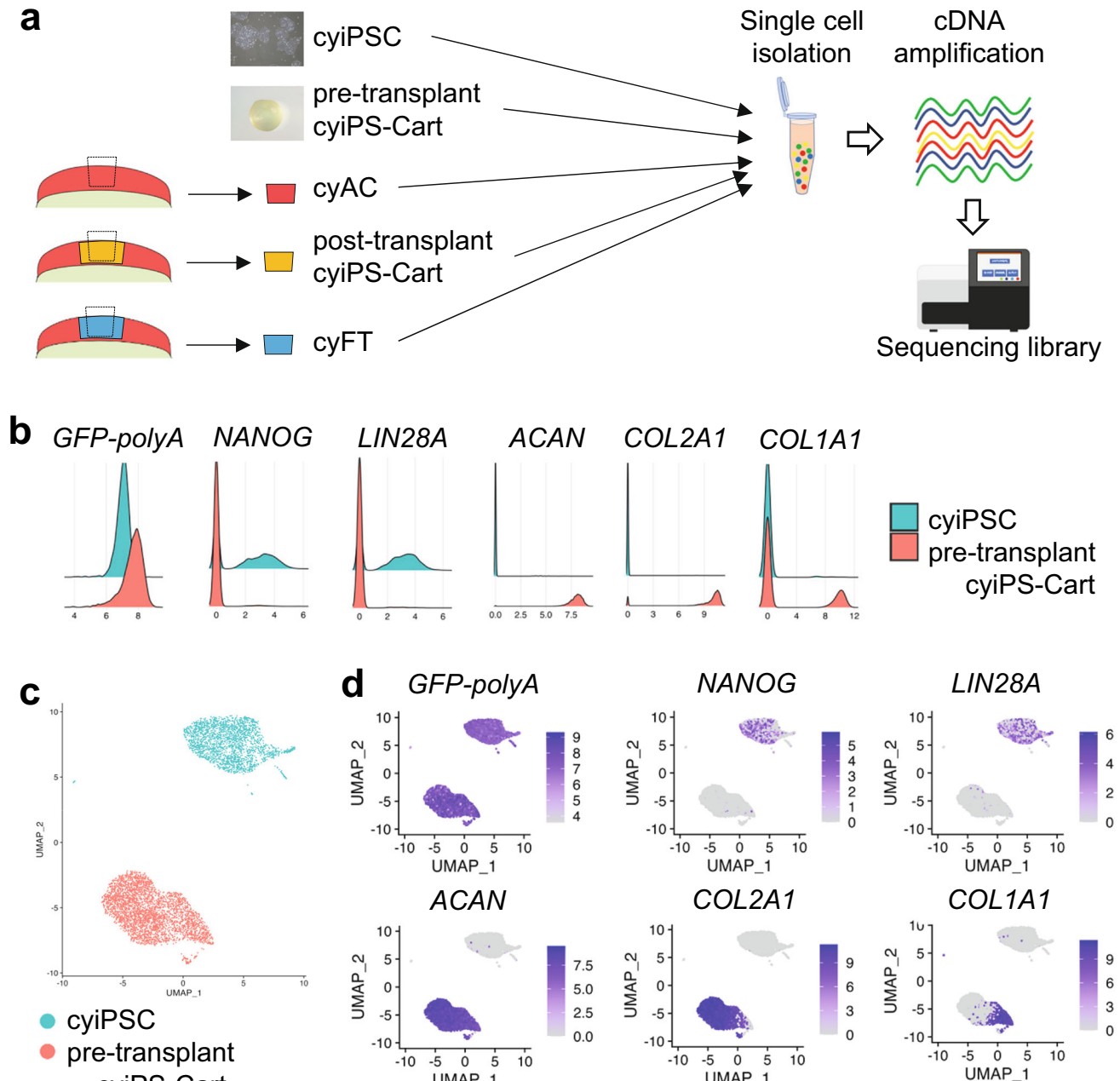

**Fig. 5 | scRNA-seq analysis and chondrogenic differentiation of cyiPSCs into pre-transplant cyiPS-Cart. a** Schematic representation of samples subjected to scRNA-seq analysis. Undifferentiated cyiPSCs (cyiPSC), cyiPS-Cart (pre-transplant cyiPS-Cart), intact articular cartilage (cyAC), fibrous tissue formed in chondral defects in the empty group (cyFT), and cyiPS-Cart in chondral defects in the transplantation group (post-transplant cyiPS-Cart) 17 weeks after surgery. **b** Ridgeplot (Seurat) showing the distribution of single-cell gene expression in each sample. The x-axis of each panel represents the expression levels of the indicated genes. The y-axis represents the number of cells. **c** CyiPSCs and pre-transplant cyiPS-Cart cells were projected onto UMAP plots with a parameter resolution of 0.5. **d** Marker gene expression levels are indicated in each cell projected on the UMAP plot using the featureplot function.

the DEGs to Ingenuity Pathway Analysis (IPA, Qiagen). IPA detected possible upstream regulators, including activated TGF-β1, activated TGF-β3, activated SMAD3, activated TGF-β, activated TGF-β2, and inhibited SMAD7 (Supplementary Data 2). The addition of TGF-β upregulates *Prg4* expression in chondrocytes[20,21]. The addition of TGF-β1 to the culture of cyiPS-Cart cells increased *PRG4* mRNA expression (Fig. 7e), whereas TGF-β inhibitor downregulated *PRG4* mRNA expression (Fig. 7e). These results suggest that the TGF-β signaling pathway is involved in *PRG4* activation in cyiPS-Cart after transplantation.

IPA also detected forskolin as an upstream activator of genes whose expression was upregulated in PRG4-positive cells in cyiPS-Cart

(Supplementary Data 2). Forskolin upregulates *PRG4* expression in chondrocytes by increasing the concentration of cAMP, which activates PKA and CREB[22]. In contrast, forskolin inactivates salt-inducible kinase 2 (SIK2) via the PKA-dependent phosphorylation of SIK2 in hepatocytes[23]. Thus, we hypothesized that SIK is involved in regulating *PRG4* expression and performed further experiments. Among the SIK family members, SIK3 mainly functions in chondrocytes[24–26]. Forskolin increased the amount of Sik3 phosphorylated at threonine 411 (pSIK3 (pT411), an inactive form of Sik3 (Fig. 8a)) and increased *Prg4* expression (Fig. 8b) in murine chondrocytes, indicating an association between Sik3 activity and *Prg4* expression. *Prg4* mRNA expression increased in primary

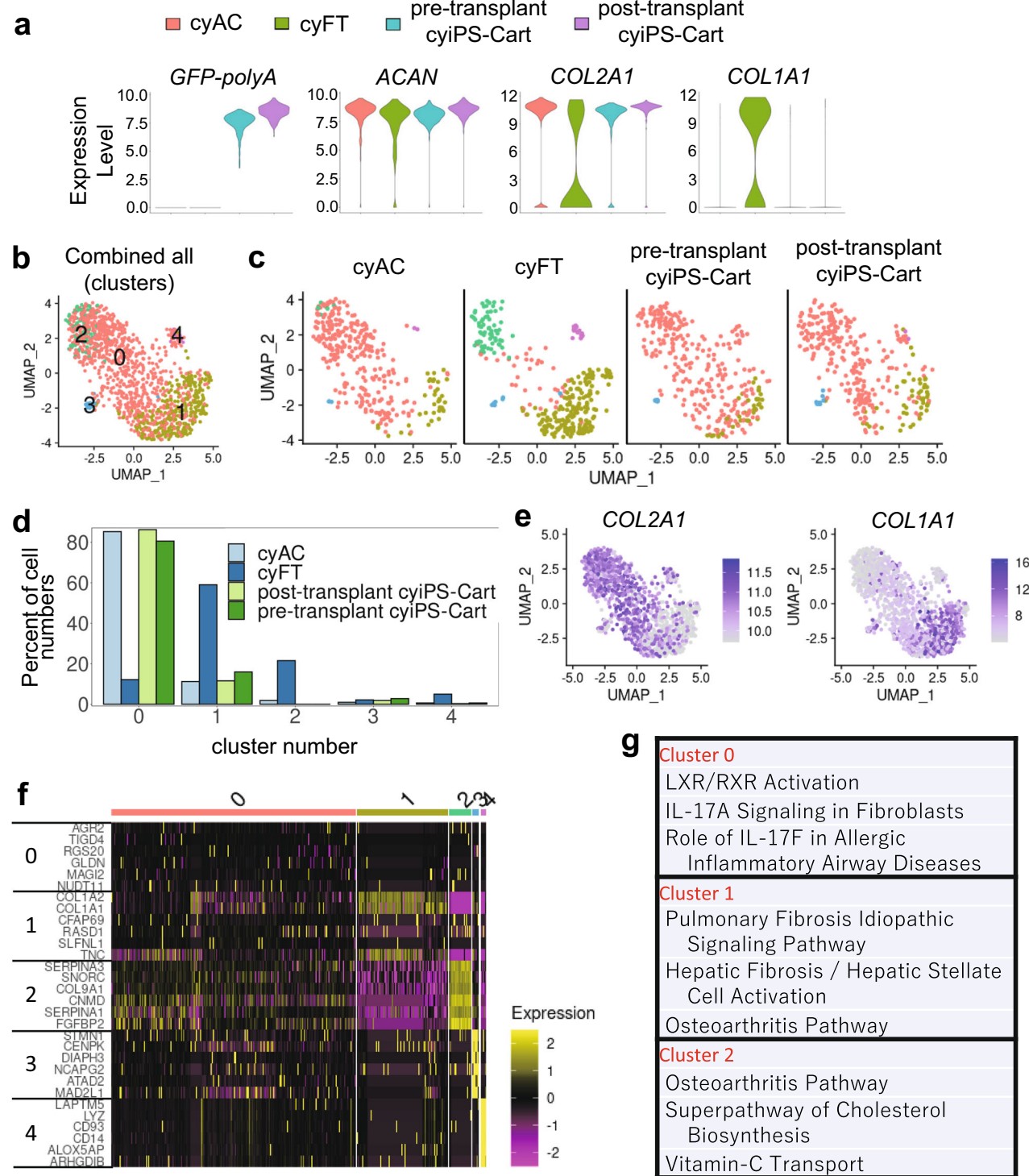

**Fig. 6 | scRNA-seq analysis of cyAC, cyFT, pre-transplant cyiPS-Cart, and post-transplant cyiPS-Cart.** **a** The VlnPlot (Seurat) shows the distribution of single-cell gene expression in each sample. The y-axis of each panel represents the expression levels of the indicated genes. **b** After reducing the cell number for each sample to 320, the data from the samples were integrated. The cells were then clustered with a parameter resolution of 0.2 and projected onto the UMAP plots. **c** UMAP plot in (**b**) separated by samples. **d** The ratio of the number of cells in each cell cluster in each sample (**c**) is plotted. **e** *COL2A1* and *COL1A1* expression levels are indicated in each cell projected on the UMAP plot using the featureplot function. **f** Heatmap revealing the scaled expression of differentially expressed genes for each cluster defined in (**b**). **g** Canonical pathways enriched for each cluster based on differentially expressed genes. The results of Clusters #3 and #4 were omitted because there were very few cells in these clusters.

chondrocytes obtained from Sik3 knockout mice but decreased in primary chondrocytes obtained from transgenic mice overexpressing *Sik3* in chondrocytes (Fig. 8c). Immunohistochemical analysis showed that the population of chondrocytes expressing Prg4 increased in Sik3

conditional knockout mice (Fig. 8d and Supplementary Fig. 7a). These results suggested that Sik3 inhibits *Prg4* expression. In vivo, the joint surface is subjected to fluid flow shear stress (FFSS), inducing the expression of the *Prg4* gene[22]. To mimic this in vivo situation, FFSS was

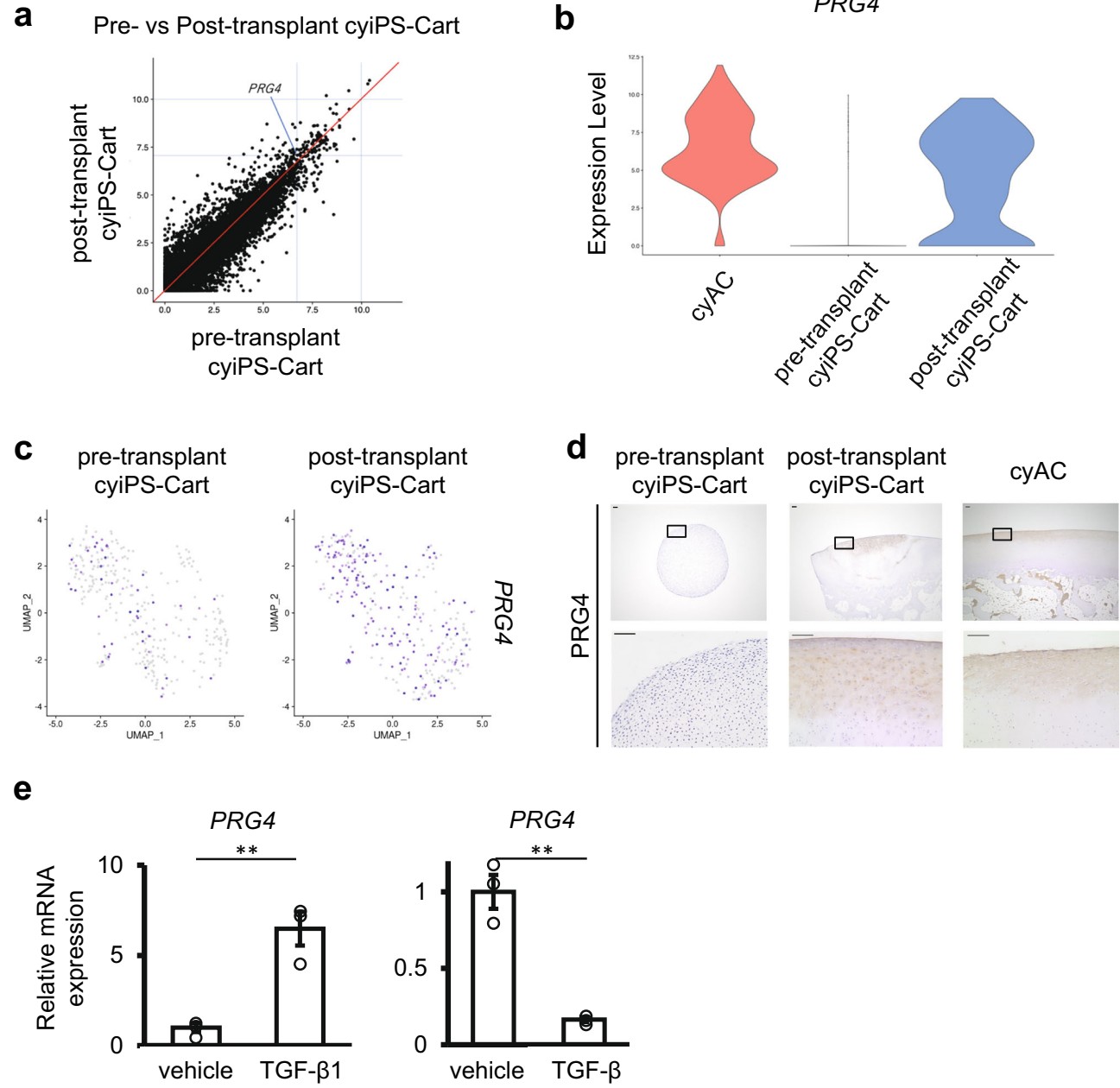

**Fig. 7 | Expression of *PRG4* in cyAC, pre-transplant cyiPS-Cart, and post-transplant cyiPS-Cart. a** The expression level of each gene in pre-transplant cyiPS-Cart is plotted on the x-axis and the expression level in post-transplant cyiPS-Cart is plotted on the y-axis. **b** VlnPlot of *PRG4* expression for each sample. **c** *PRG4* expression levels indicated in each cell projected on the UMAP plot in Fig. 6b, using the FeaturePlot function. **d** Histological sections were immunostained for PRG4 expression. A magnification of the boxed regions in the top row is shown in the bottom row. Data were representative of three cyiPS-Cart organoids and three monkeys. Scale bars, 100 μm. **e** Cells from the pre-transplant cyiPS-Cart were cultured in the presence or absence of TGF-β1 (*left*) or TGF-β inhibitor, SB431542 (right). *PRG4* mRNA expression was analyzed using real-time RT-PCR. Error bars denote means ± SE. **$P = 0.0048$, **$P = 0.0017$ by two-tailed Student's *t*-test ($n = 3$ dishes). Data were representative of three independent experiments. Source data are provided as a Source Data file.

applied to mouse primary chondrocytes (Supplementary Fig. 7b). Application of FFSS to wild-type mouse primary chondrocytes increased the expression of *Prg4* after 12 h (Fig. 8e and Supplementary Fig. 5c, open circles). *Sik3* deletion further increased FFSS-induced *Prg4* expression (Fig. 8e and Supplementary Fig. 7c, closed circles), suggesting that Sik3 helps regulate *Prg4* expression in mouse chondrocytes. The induction of *PRG4* indicates that the post-transplant cyiPS-Cart acquired lubrication function as articular cartilage.

We analyzed the relationship between TGF-β and Sik3 inactivation. The addition of TGF-β did not affect the phosphorylation of Sik3 at T411 (Supplementary Fig. 8), whereas the addition of forskolin increased the phosphorylation of Sik3 at T411 but did not affect the phosphorylation of Smad3 (Supplementary Fig. 8). These results suggest that TGF-β and Sik3 regulate *Prg4* expression independently.

The restricted expression of PRG4 in the superficial zone of the post-transplant cyiPS-Cart suggests that cyiPS-Cart after transplantation survived and directly contributed to the repair of tissue in chondral defects, and also functioned as articular cartilage. This is further supported by the result that the transplantation of cyiPS-Cart prevented the degeneration of the native articular cartilage surrounding

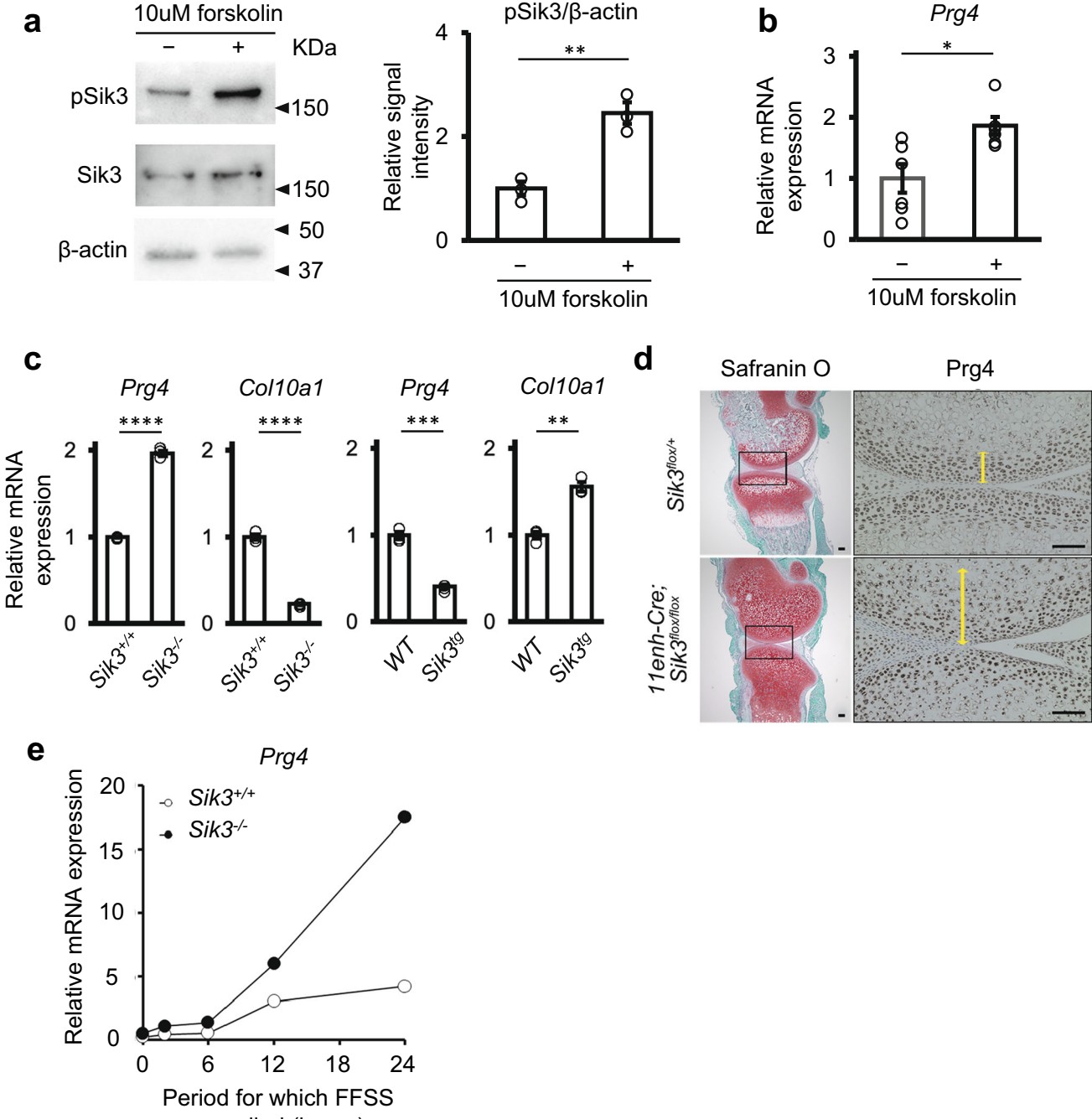

**Fig. 8 | Involvement of Sik3 function and fluid flow shear stress (FFSS) on *Prg4* expression in mouse chondrocytes. a** Immunoblot expression analysis of Sik3 phosphorylated at T411, an inactive form of Sik3, in wild-type mouse primary chondrocytes treated with 10 μM forskolin for 30 min. Left: blots representative of three independent experiments are shown. Right: quantification of pSik3(pT411) to the β-actin ratio in cultured chondrocytes with or without forskolin treatment. Error bars denote mean ± SE. **$P$ = 0.0041 by two-tailed Student's *t*-test ($n$ = 3). **b** Real-time RT-PCR analysis of *Prg4* expression in wild-type mouse primary chondrocytes treated with 10 μM forskolin for 6 h. Error bars denote mean ± SE. *$P$ = 0.0108 by two-tailed Student's *t*-test ($n$ = 6 dishes). Data were representative of three independent experiments. **c** Real-time RT-PCR analysis of *Prg4* and *Col10* expression in primary chondrocytes obtained from Sik3 knockout (*Sik3*$^{-/-}$) and Sik3 transgenic (*Sik3*$^{tg}$) mice. Error bars denote mean ± SE. ****$P$ < 0.0001, ****$P$ < 0.0001, ***$P$ = 0.0003, **$P$ = 0.0012, $n$ = 3, two-tailed Student's *t*-test ($n$ = 3 dishes). Data were representative of two independent experiments. **d** Immunohistochemical analysis of Prg4 expression in the knee joints of *Sik3* conditional knockout (*11Enh-Cre; Sik3*$^{flox/flox}$) mice lacking Sik3 expression in chondrocytes 14 days after birth. Yellow arrows indicate the thickness of the area in which Prg4 was expressed. Data were representative of five conditional knockout mice and four *Sik3*$^{flox/+}$ mice. Scale bars: 100 μm. **e** Real-time PCR analysis of Prg4 expression in wild-type (*Sik3*$^{+/+}$) and *Sik3* knockout (*Sik3*$^{-/-}$) primary chondrocytes subjected to FFSS for the indicated period. Data are representative of two independent experiments. Source data are provided as a Source Data file.

chondral defects. These results collectively indicate the successful engraftment of allogeneic cyiPS-Cart in a chondral defect in the knee joints of the primate model.

## Discussion

To the best of our knowledge, this is the first study to provide scientific evidence in a primate model that allogeneic cartilage achieves engraftment in chondral defects, without inducing immune reactions. In contrast, allogeneic cyiPS-Cart elicits an immune reaction in osteo-chondral defects[14]. Cartilage is believed to be relatively immunoprivileged[6,27] because chondrocytes are surrounded by the ECM, which protects the chondrocytes from exposure to cells involved in immunological reactions. The lack of an immune reaction following cyiPS-Cart transplantation in chondral defects, which bleed minimally, suggests that exposure of allogeneic cyiPS-Cart in osteochondral defects to abundant blood flow in the bone marrow might elicit an immune reaction despite the presence of ECM. Although allogeneic cyiPS-Cart in osteochondral defects survives, further studies are needed to determine whether the degree of the immune reaction is tolerable for effective regeneration. While it remains to be determined whether osteochondral defects can be cured by allogeneic cartilage transplantation[28], our results indicate that chondral defects can be a definite indication for allogeneic cartilage transplantation.

Most cell-based therapies induce cartilage regeneration through a mechanism called trophic effects, wherein implanted cells survive only transiently and secrete growth factors that stimulate the host progenitor cells. Evidence that implanted cells achieve engraftment and directly contribute to tissue repair is scant[29–31]. In a recent study, human embryonic stem cell (ESC)-derived chondrocytes that do not associate with cartilage ECM were implanted into the articular cartilage defects of mini-pigs[32]. Of the cells that formed in the repaired tissue in the defects, 4% were human, indicating that most of the cells that form in repaired tissue are those of the recipient. Although experimental conditions of the two studies differ (xenograft vs. allograft; critical vs. small size defects; 6 vs. 4 months' observation), allogeneic cyiPS-Cart survived for at least four months, and almost all cells in the repair tissue were transplanted cells, as demonstrated by GFP expression in our study. In addition, cyiPS-Cart cells differentiated into articular chondrocytes after transplantation and functioned as articular cartilage. Our results suggest that SIK3 could be involved in post-transplant differentiation. Although we do not know whether this differentiation mechanism is specific to cyiPS-Cart, we speculate that two specific features of iPS-Cart could favor its cells to differentiate and survive: First, the iPS-Cart is composed of chondrocytes and cartilage ECM, which contribute to the survival and differentiation of chondrocytes by providing an appropriate environment for chondrocytes. Second, iPS-Cart has characteristics of embryonic cartilage[33] that would contribute to its survival and differentiation. The survival and differentiation capacity of cyiPS-Cart enables a new strategy whereby damaged cartilage can be replaced with transplanted cartilage. Regarding clinical relevance, it has not been known whether engraftment of cartilage transplants gives better clinical results, such as improved joint function and pain relief, than repair tissue formed by trophic effects or vice versa. It is plausible that engraftment of cartilage transplants is better indicated for severe cartilage lesions where the provision of host progenitor cells is limited. Further study is required to clarify the indications.

The treatment of chondral defects is challenging, particularly in two aspects. First, the chondral defects do not bleed and, therefore, do not initiate the wound-healing process. Therefore, chondral defects are occasionally treated by microfracture surgery, wherein the subchondral bone is invasively pierced to introduce mesenchymal cells into the defect. Cell-based therapies combined with microfracture probably initiate a regenerative mechanism by which implanted cells secrete factors that stimulate mesenchymal cells in the bone marrow, which in turn improves tissue repair. However, the repair tissue remains fibrous because the chondrogenic capacity of host progenitor cells is limited. The regenerative mechanism of survival and replacement by cyiPS-Cart does not require an invasive procedure for microfracture because it heals the defect independently of host progenitor cells.

Second, the integration between the repaired tissue and the surrounding native cartilage is hardly achieved[1,34,35]. Thus, the integration of cyiPS-Cart with host cartilage is promising. Human iPSC-derived cartilage organoids have the capacity for integration[36]. FGF signals are involved in this integration. Further studies are needed to fully understand this integration mechanism and to enhance the integration of cartilage grafts.

There are several limitations to this study. First, the *cynomolgus monkeys* used in this study are not large animals, which makes it difficult to reproduce changes in biomechanics observed in cartilage lesions in humans. Although chondral defects in the empty group were filled with fibrous tissue in a four-month interval, the small size of the defects would eventually heal after a longer period. Hence, our results will need to be confirmed in larger animal models with critical-sized defects. In addition, we did not analyze the biomechanical properties of post-transplant cyiPS-Cart. A biomechanical test will be required to determine clinical relevance. Second, an observation period of 17 weeks (4 months), as employed in this study, is not long enough to demonstrate the sustainability of a transplant. It is, however, still possible to conclude that engrafted cells had obtained a degree of articular chondrocyte identity by 4 months. A longer observation period, such as 2 years, would further signify the sustainability and turnover of transplants. Third, although our results suggest that Sik3 inhibits *Prg4* expression in mouse chondrocytes, this does not prove that SIK3 is involved in *PRG4* expression in post-transplant cyiPS-Cart in monkeys. In conclusion, cyiPS-Cart transplanted into chondral defects survived, integrated with native cartilage, acquired *PRG4* expression in the superficial region, and prevented degeneration of the surrounding cartilage. These results collectively suggest that allogeneic cyiPS-Cart engraftment will contribute to the development of translational medical techniques based on allogeneic pluripotent stem cells to treat chondral defects in articular cartilage.

## Methods
### Ethics statement
All methods were performed following relevant guidelines and regulations. Experiments using recombinant DNA were approved by the Recombinant DNA Experiments Safety Committees of Kyoto University (No. 180041) and Osaka University (No. 04794). All animal experiments were approved by the Institutional Animal Committees of Kyoto University (No. 18-101-14 and No. 16-74-17) and Osaka University (No. 03-044-014).

### Isolation of *cynomolgus monkey* iPSCs and creation of cyiPS-Cart by chondrogenic differentiation of cyiPSCs
We prepared a cyiPSC line, 1466A1, in which the EGFP gene was integrated into the AAVS1 locus using the pBS-macAAVS1-P-CAG-GFP vector and CRISPR-Cas9 system. The cyiPSC lines had homozygous MHC haplotypes (Mafa-HT1; Mafa is the MHC of a *cynomolgus macaque*)[37].

The cyiPSCs were maintained on a mitomycin C-inactivated feeder layer of mouse embryonic fibroblasts (MEF) in Dulbecco's Modified Eagle Medium/ Ham's F12 (DMEM/F12) medium (Sigma) containing 20% knockout serum replacement (KSR; Thermo Fisher Scientific), $1 \times 10^{-4}$ M 2-mercaptoethanol (Thermo Fisher Scientific), $1 \times 10^{-4}$ M nonessential amino acids (Thermo Fisher Scientific), 1 mM sodium pyruvate (Thermo Fisher Scientific), 2 mM GlutaMAX (Thermo Fisher Scientific), and 50 units penicillin and 50 mg/mL streptomycin (1% PC/SM, Thermo Fisher Scientific). The medium was changed every day. Every 7 days, the cyiPSC colonies were subjected to 0.1% collagenase treatment for 3 min

(Thermo Fisher Scientific), and cells were collected, centrifuged, and replated onto dishes with new MEF feeder cells[38].

For chondrogenic differentiation, all cyiPSCs were loosely detached from the MEF feeder cells by exposure to 0.1% collagenase and cultured in Stemfit AK02N medium (Ajinomoto, Tokyo, Japan) on Matrigel-coated dishes for 7 days. CyiPSCs were differentiated into chondrocytes, and cartilage was formed using a previously described method for human iPSCs[12,13]. Briefly, after chondrogenic differentiation, the cells were transferred into suspension culture to form cartilaginous particles 1–3 mm in diameter. Differentiated chondrogenic cells and particles were cultured in a chondrogenic medium (DMEM [Sigma] with 1% ITS-X [Thermo Fisher Scientific], 1% FBS [Thermo Fisher Scientific], $1 \times 10^{-4}$ M nonessential amino acids, 1 mM sodium pyruvate, 1% PC/SM, 50 µg/mL ascorbic acid [Nacalai Tesque], 10 ng/mL BMP2 [Peprotech], 10 ng/mL TGFβ1 [Peprotech], and 10 ng/mL GDF5 [BioVision]). We also created cyiPS-Cart from the previously reported cyiPSC line, 1231C1-G[39].

## Transplantation of cyiPSC-derived cartilage organoid into osteochondral or chondral defects in *cynomolgus monkey*

*Cynomolgus monkeys* (3–4 years old) were purchased from Ina Research (Nagano, Japan). Under general anesthesia, the skin and joint capsules of the right knee were opened in 12 monkeys. Chondral defects (1 mm diameter and 0.5 mm depth) were created at the trochlea of the distal femur under surgical microscopy. We used the electric router (Dremel, micro, please see figure below) and a dental steel bur under a surgical loupe to create consistent chondral defects. We transplanted 1466A1 cyiPSC-derived cyiPS-Cart into the chondral defects by press-fitting in six monkeys (transplantation group). We transplanted nothing into the other six monkeys (empty group). The joint capsule and the skin were closed. After surgery, we intramuscularly injected antibiotics and buprenorphine hydrochloride (0.1 mg/body) for 3 days into all monkeys.

Three monkeys in each group were sacrificed either 4 or 17 weeks after injecting pentobarbital sodium (100 mg/kg) under deep anesthesia. For each monkey, we harvested and subjected one transplanted site for histological analysis and harvested two transplanted sites and combined them for scRNA-seq analysis.

As a positive control for immune reactions, we created osteochondral defects (1.5 mm depth) in three monkeys. We transplanted 1466A1 cyiPSC-derived cyiPS-Cart into one monkey and 1231C1-G cyiPSC-derived cyiPS-Cart into two monkeys. We sacrificed the monkeys four weeks later, harvested the transplanted sites, and subjected them to histological analysis.

## CT of monkey knee joints

CT of the monkey knee joints was performed using a CT system (Aquilion TSX-101A/NA; TOSHIBA, Japan). Three-dimensional images were constructed using image processing software (Aquilion TSX-101A/NA, TOSHIBA, Japan).

## MHC genotyping of monkeys

MHC genotyping of monkeys was performed based on the MHC allele information registered in the Immuno Polymorphism Database (http://www.ebi.ac.uk/ipd/index.html) by Ina Research[37].

## Histological analysis

Samples were fixed with 4% paraformaldehyde, decalcified with KCX (FALMA), processed, and embedded in paraffin. Semi-serial sections were stained with HE or safranin O. To assess the quality of the repaired tissue, the safranin O-positive area and the total area of the repaired tissue were measured by BZ-X800 analyzer software (Keyence Corp., Japan), and the former was divided by the latter.

The safranin O-positive area and the total area of articular cartilage regions surrounding chondral defects were measured to assess the quality of articular cartilage surrounding chondral defects; the former was divided by the latter.

Two individual assessors reviewed the sections in a blinded manner and scored them according to a modified Wakitani histological scoring system[40,41]. The maximum score of this system is 11, and a lower score indicates repair more similar to the native articular cartilage (Supplementary Table 2)[40].

Semi-serial sections were immunostained using specific antibodies. Supplementary Table 3 lists the antibodies used in this study. Anti-type I collagen, anti-type II collagen, anti-GFP, and anti-PRG4 antibodies were detected using a CSA II Biotin-free Tyramide Signal Amplification System Kit (Agilent Technologies, Santa Clara, CA, USA) and DAB was used as the chromogen. For the anti-CD3 antibodies, immune complexes were detected using secondary antibodies conjugated to Alexa Fluor 488. The antigens were unmasked by treatment with hyaluronidase and EDTA.

The number of positively stained cells with anti-CD3 antibodies in the region below the osteochondral junction was counted in four fields per knee.

## Preparation of single cells for scRNA-seq analysis

The cyiPS-Cart (pre-transplant cyiPS-Cart), intact articular cartilage (cyAC), fibrous tissue formed in chondral defects in the empty group (cyFT), and cyiPS-Cart in chondral defects in the transplantation group (post-transplant cyiPS-Cart) were minced into 1–2 mm pieces. Next, we digested these pieces with Liberase solution (for cyAC, and post-transplant cyiPS-Cart: RPMI-1640 (Nacalai Tesque) supplemented with 0.2% FBS, 10 mM HEPES pH 7.2–7.4, 0.2–0.4 mg/mL Liberase TM (Roche), and 2 kU/mL DNase I (Merck); for pre-transplant cyiPS-Cart: DMEM with 1% FBS, 1% ITS-X, 50 µg/mL ascorbic acid, 1 mM sodium pyruvate, 1% nonessential amino acids, 1% penicillin-streptomycin, 10 ng/mL TGF-β1, 10 ng/mL GDF5, 10 ng/mL BMP2, 0.2 mg/mL Liberase TM (Roche), and 2 kU/mL DNase I (Merck)) at 37 °C, 5% $CO_2$ for 120–210 min with continuous shaking. The cells were mixed ten times using a 1000 µL blue tip-fitted pipette and then passed through a cell strainer (70 µm pore size; BD Biosciences), centrifuged at 4 °C for $300 \times g$ for 5 min, and the supernatant was discarded. The cells were resuspended in RPMI-1640 medium supplemented with 0.2% FBS and 10 mM HEPES.

For cell hashing, we biotinylated cell surface proteins using EZ-Link Sulfo-NHS-Biotin (Thermo Scientific), and then stained them with 0.6 µg/mL Totalseq (A0951-A0955, and A0436 (BioLegend))[42]. We selected live cells stained with A0951-A0955 using a FACS Aria II flow cytometer (BD Biosciences) and suspended them in a sample buffer (BD Biosciences).

## cDNA synthesis using the BD Rhapsody system

We subjected the obtained single-cell suspensions to a BD Rhapsody system using the BD Rhapsody Targeted & Abseq Reagent kit (BD Biosciences). After reverse transcription, the BD Rhapsody beads were treated with exonuclease I at 37 °C for 60 min and 1200 rpm on a Thermomixer C with a Thermotop. The resultant beads were immediately chilled on ice. The supernatant was removed, and the beads were washed with 1 mL WTA wash buffer (10 mM Tris-HCl pH 8.0, 50 mM NaCl, 1 mM EDTA, and 0.02% Tween-20) and 200 µL BD Rhapsody lysis buffer (for inactivation of enzyme), once again with 1 mL WTA wash buffer alone, twice with 500 µL WTA wash buffer, and finally resuspended in 200 µL bead resuspension buffer and stored at 4 °C. During the washing step, bead-containing DNA LoBind tubes were replaced twice.

## Generation of TAS-Seq library

TAS-Seq libraries were generated by Immunogeneteqs Inc. (Noda City, Chiba, Japan), as described previously[43]. Briefly, reverse-transcribed exonuclease I-treated BD Rhapsody beads were subjected to a

terminator-assisted TdT reaction, second-strand synthesis reaction, and a first/second round of whole-transcriptome amplification (WTA) or Totalseq library amplification reaction. The size distribution and concentration of the amplified cDNA and hashtag libraries were analyzed using a MultiNA system (Shimadzu). Illumina libraries were constructed from amplified cDNA libraries (100 ng) using the NEBNext Ultra II FS Library Prep kit for Illumina (New England Biolabs). Illumina adapters and unique-dual barcodes were added to the hashtag libraries using PCR. The size distribution and concentration of the amplified Illumina libraries were analyzed using the MultiNA system and KAPA library quantification kit (KAPA Biosystems). Sequencing was performed using an Illumina Novaseq 6000 sequencer (Illumina, San Diego, CA, USA) and a Novaseq 6000 S4 reagent kit v1.0 or v1.5, according to the manufacturer's instructions (read 1 (cell barcode): 67 bp and read 2 (cDNA): 140 (v1.0) / 155 (1.5) base-pair with 8 base-pair ×2 unique-dual indexes). The pooled library concentration was adjusted to 1.75 nM (v1.0) or 2.0 nM (v1.5), and the library was spiked with 12% PhiX control library v3 (Illumina).

### Fastq data preprocessing and generation of the single-cell gene expression matrix

To obtain the gene expression count matrix and Totalseq expression matrix for every cell, fastq files of the TAS-Seq data were processed by Immunogeneteqs Inc. as described previously[43]. Bowtie2-indexes built from reference RNA sequences (cDNA and ncRNA FASTA files from the Ensembl database (*Macaca fascicularis*_6.0)[44]) were used to assign cDNA reads to each transcript. Associated Totalseq streptavidin/anti-biotin reads were mapped to known barcode sequences (provided by BioLegend) using bowtie2-2.4.2, with the following parameters: -p 2 -D 20 -R 3 -N 0 -L 8 -i S,1,0.75 -norc -seed 656565- reorder-trim-to3:39 -score-min L,−9,0 -mp 3,3 -np 3 -rdg 3,3. The inflection point of the knee plot (total read count versus the rank of the read count) was detected using the DropletUtils package[45] in R 3.6.3 (https://cran.r-project.org/) from the resulting single-cell gene expression matrix files. Cells for which the total read count exceeded the inflection point were considered valid. Demultiplexing of single cells by expression of Totalseq streptavidin/anti-biotin was performed as described previously[43].

### Background subtraction of TAS-Seq expression matrix by distribution-based error correction

To reduce the background read counts of each gene possibly derived from RNA diffusion during the cell lysis step within the BD Rhapsody cartridge and reverse transcription, a distribution-based error correction was included in the BD Rhapsody targeted scRNA-seq workflow performed by Immunogeneteqs Inc., as previously reported[43]. Briefly, the genes for which the $\log_2(x+1)$-transformed maximum expression was over 8 were selected, and a biexponential transformation was applied to each gene count using the FlowTrans package[46] in R 3.6.3. Next, Gaussian mixture components were detected using the mclust package[47] in R 3.6.3. The average expression of each component was calculated, and the genes for which the maximum average expression of each component was over 5.5 were selected. If the difference between the average expression of each component and its maximum expression was greater than 5, the expression level of the components was considered as background gene expression, and the converted expression of the components was set to 0.

### Single-cell clustering and annotation

We clustered single cells from each dataset using Seurat v4.0.3[48] in R 4.1.0. The Seurat object for each dataset was created using the CreateSeuratObject function (min. cells = 5, min. genes = 500). Cells for which the percentage of mitochondrial genes was greater than the threshold were filtered using the subset function in Seurat v4.0. The expression data were normalized using the NormalizeData function

(scale factor = 1,000,000, according to the analytical parameter used by Muris[49]). Cells were categorized into the S, G1, or G2/M phases by scoring cell cycle-associated gene expression using the CellCycleScoring function. Highly variable genes in each dataset were identified using the FindVariableFeatures function with the following parameters: selection.method = "vst," nfeatures = 5000, mean.cutoff = c(0.1, Inf), and dispersion.cutoff = c(0.5, Inf). The expression data were scaled using the ScaleData function. The read counts of each cell within each dataset were regressed as confounding factors in the ScaleData function. Principal component analysis (PCA) was performed using the RunPCA function, and the top 42 PCs were selected for dimensional reduction using UMAP. We determined the cluster resolution of the values. DEGs of each cluster were determined using the FindMarkers function with a 0.05 p_val_adj threshold between the two groups (i.e., cell clusters). DEGs were further analyzed by IPA v01-20-04 (QIAGEN) to identify upstream genes and pathways.

### RNA velocity analysis

We performed the RNA velocity analysis as described previously[50]. TAS-Seq data cDNA reads were mapped to the reference genome (*Macaca fascicularis*_6.0 for *macaca fascicularis* data, and GRCh38 release-101 for human data) using HISAT2-2.2.1[51] and the following parameters were used: -q -p 6 −rna-strandness F −very-sensitive −seed 656565 −reorder −omit-sec-seq −mm. For the HISAT2 index build, a corresponding ensembl gtf file was filtered to retain protein-coding RNA, long non-coding RNA, and T cell chain/immunoglobulin chain annotations according to the 10X Genomics's method (https://support.10xgenomics.com/single-cell-gene-expression/software/pipelines/latest/advanced/references#mkgtf). Then, the cell barcode information of each read was added to the HISAT2-mapped BAM files, and associated gene annotations were assigned using featureCounts v2.0.2[52] with the following parameters: -T 2 -Q 0 -s 1 -t gene -g gene_-name −primary -M -O −largestOverlap −fraction -R BAM. In the featureCounts analysis, a "gene" annotation was used to capture unspliced RNA information for the RNA velocity analysis, and primary annotations were kept. The resulting BAM file was split using valid cell barcodes and nim 1.0.6 and hts-nim v0.2.23, the split files were processed into loom files using velocyto run (version 0.17.17) with the -c and -U options, and the loom files were concatenated using the loompy.combine function (version 3.0.6)[53]. Then, we used scVelo[54] for RNA velocity analysis. The loom files were read to an AnnData object. After estimating the RNA velocity, we inferred the trajectory using PAGA[55]. The velocity-inferred directionality extended the trajectory.

### Isolation and culture of cells from pre-transplant cyiPS-Cart

The pre-transplant cyiPS-Cart was treated with 0.25% trypsin-EDTA (Thermo Fisher Scientific) for 1 h and subsequently treated overnight with 4 mg/mL collagenase D (Roche) in DMEM supplemented with 1% PC/SM. After washing, cells were suspended in DMEM supplemented with 10% FBS or chondrogenic medium.

A total of $3.0 \times 10^5$ cells from pre-transplant cyiP-Cart were plated in a ɸ 35 mm dish and cultured in DMEM supplemented with 10% FBS in the presence or absence of 100 ng/mL TGF-β1 (Peprotech). A total of $2.3 \times 10^5$ cells from pre-transplant cyiPS-Cart were cultured in a chondrogenic medium in the presence or absence of 100 μM TGF-β inhibitor SB431542 (Cayman). The cells were collected 48 h later for mRNA expression analysis.

### Conventional and conditional *Sik3* knockout mice and *Sik3* transgenic mice

All mice used were C57BL/6. *Sik3*$^{-/-}$ mice[25] and *Sik3*$^{flox/flox}$ mice[24] have been previously described. *11Enh-Cre* mice are transgenic mice expressing Cre under the control of *Col11a2* promoter/enhancer sequences[56]. *11Enh-Cre* transgenic mice and *Sik3*$^{flox/flox}$ mice were mated to generate *Sik3* conditional knockout mice, in which Sik3 was

specifically deficient in chondrocytes. *Sik3*⁻/⁻ and *11Enh-Cre; Sik3*^*flox/flox* mice exhibited similar cartilage phenotypes. *Col11a2-hSIK3* transgenic mice overexpressing human SIK3 in chondrocytes under the *Col11a2* promoter/enhancer control were described previously[25].

## Culture of mouse primary chondrocytes

Primary chondrocytes were prepared from *Sik3*⁻/⁻ knockout mice at 18.5 postcoitus and *Col11a2-hSIK3* transgenic mice 3 days after birth, as described previously[57]. Briefly, epiphyseal cartilage was dissected from the knee, elbow, shoulder joints, and femoral heads of mice and digested with 3 mg/mL collagenase D (Roche) in DMEM/F12 (Invitrogen) containing 5% FBS and 1% penicillin-streptomycin (Life Technologies) at 37 °C overnight. Approximately $5 \times 10^5$ primary chondrocytes were obtained from each mouse and cryopreserved in LaboBanker (Kurabo Industries Ltd.). Before the experiments, the cells were lysed, plated, and cultured in DMEM/F12 supplemented with 5% FBS, and 1% penicillin-streptomycin for less than 10 days.

Chondrocytes were seeded ($0.5 \times 10^5$ cells/well) into 12-well tissue culture plates (Corning). Chondrocytes were cultured in DMEM/F12 containing 5% FBS and 1% penicillin-streptomycin (Invitrogen) at 5% $CO_2$ in humidified air. Before experiments, cells were pretreated overnight with a starvation medium (serum-free). For experiments, cells were treated with 1, 5, or 10 ng/mL TGF-β1 (PeproTech) or 10 or 100 µM forskolin (F3917, Sigma-Aldrich) (in DMSO) for 30 min or 1 h for immunoblot analysis and 6 h for mRNA expression analysis.

## Application of FFSS

We followed a previously described method[22]. Mouse primary chondrocytes were seeded ($1 \times 10^5$ cells/well) in six-well tissue culture plates (diameter = 3.48 cm) (Corning). Chondrocytes were cultured in DMEM/F12 containing 5% FBS and 1% penicillin-streptomycin (Invitrogen) with 5% $CO_2$ in humidified air for 2–24 h. Thirty min before exposure to static or FFSS conditions, the medium was replaced gently with 4 mL fresh culture medium. For cultures exposed to FFSS, culture dishes were placed on an orbital shaker (0.5-cm radius of gyration) (IKA, KS 260 basic) set to rotate at 300 rpm at 37 °C, which exposed the chondrocytes to a steady laminar, non-pulsatile shear stress of -7.6 dyn/cm². 

## Immunoblot analysis

Mouse primary chondrocytes were lysed in RIPA buffer (10 mM Tris-HCl pH 7.5, 150 mM NaCl, 0.1% SDS, 0.1% sodium deoxycholate, 1 mM EDTA, 1% NP-40, complete protease inhibitors from Roche, and phosphatase inhibitor cocktail 1 and 2 from Sigma-Aldrich) and subjected to SDS-PAGE.

The separated proteins were electroblotted and the membranes were immunostained with rabbit anti-SIK3 (Abcam, 1:500), rabbit anti-pSIK3 (pT411) (KINEXUS, 1:1,000), rabbit anti-Smad2/3 (CST #8685 S, 1:1000, rabbit anti-pSmad3 (CST #9520 S, 1:1000), and rabbit anti-β-actin (Cell Signaling, 1:1,000) antibodies. Goat anti-mouse IgG-HRP (Santa Cruz, 1:5000) or goat anti-rabbit IgG-HRP (Santa Cruz, 1: 5000) were used as secondary antibodies. ECL system and LAS4000 (GE Healthcare) were used for chemiluminescent immunodetection. pSIK3(pT411) levels were quantified relative to β-actin levels by using the FUSION FX software (Vilber, France). The uncropped blots are provided in Source Data and Supplementary Fig. 9.

## mRNA expression analysis

Total RNAs were extracted using RNeasy (Qiagen). For quantitative reverse transcription PCR (RT-PCR), total RNA was reverse-transcribed into first-strand cDNA using ReverTra Ace (Toyobo) and an oligo(dT) 20 primer. PCR amplification was performed using the KAPA SYBR FAST qPCR Master Mix ABI prism kit (KAPA Biosystems, Wilmington, USA) and StepOnePlus Real-Time PCR System (Thermo Fisher Scientific). The sequences of the PCR primers used are listed in Supplementary Table 4. The RNA expression levels of target genes were normalized to that of GAPDH or β-actin mRNA expression, and the results indicate the relative expression of the molecules.

## Statistical analysis

The data are shown as mean ± standard error (SE). We performed a two-tailed Student's *t*-test for parametric data and a one-way analysis of variance (ANOVA) with Tukey's HSD test for multiple comparisons. Statistical significance was set at $P < 0.05$.

## Reporting summary

Further information on research design is available in the Nature Portfolio Reporting Summary linked to this article.

## Data availability

All the original data are available upon request from the authors. The scRNA-seq datasets have been deposited in the GEO database under accession code GSE206120. Source data are provided with this paper.

## Code availability

The code used in this study is provided as Supplementary Code 1.

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

## Acknowledgements

We thank Chieko Matsuda, Masumi Sanada, Hiroki Hagizawa, and Yuya Nishijima for their assistance and helpful discussion. We thank the iPS Cell Research Fund for its research support. This study was supported by JSPS KAKENHI Grant No. 18H02923 (to N.T.) and WPI Premium Research Institute for Human Metaverse Medicine (PRIMe) (to N.T.) from the Japan Society for the Promotion of Science. This study was also supported by the Center for Clinical Application Research on Specific Disease/ Organ (type B) Grant No. 21bm0304004h0009 (to N.T.); Research Project for Practical Applications of Regenerative Medicine Grant No. 21bk0104079h0003 (to N.T.); Practical Research Project for Rare/ Intractable Diseases (step 1) Grant No. 21ek0109452h0002 (to N.T.); Core Center for iPS Cell Research Grant No. 20bm0104001h0008 (to N.T.); and the Acceleration Program for Intractable Diseases Research utilizing disease-specific iPS cells Grant No. 20bm0804006h0004 (to N.T.) from the Japan Agency for Medical Research and Development (AMED).

## Author contributions

K.A., S.M., and N.T. designed experiments. K.O. prepared cyiPSCs. A.Y. created the cyiPS-cart. K.A. transplanted cyiPS-Carts into monkeys. K.A. performed CT and histological analyses. K.A., M.M., S.K., and N.T. performed scRNA-seq analysis. N.H. and Y.T. performed experiments regarding Sik3. K.A. and N.T. wrote the manuscript.

## Competing interests

N.T. is an inventor and Kyoto University is a holder of the patent on "An efficient chondrocyte induction method" (PCT/JP2014/079117). This patent is licensed to Asahi KASEI corporation. Y.T. is an employee of Asahi KASEI. The remaining authors declare no competing interests.
