## [Peer Review File · Nature Communications]

Engraftment of allogeneic iPS cell-derived cartilage organoid in a primate model of articular cartilage defectREVIEWER COMMENTS

Reviewer #1 (Remarks to the Author):

In this study, Abe et al. analyzed the allogenic transplantation of MHC-mismatched iPSC cell-derived cartilage organoids in a primate animal model without the use of immunosuppressive drugs. The reviewer would like to acknowledge a significant amount and complexity of the current study. Several strengths include novelty of primate iPSC in cartilage field, large animal model of a cartilage lesion and post-implantation Single-cell sequencing of the engrafted tissue. Presented data clearly demonstrate engraftment of the implanted cells, their chondrogenic identity and lack of significant immune response. In addition, lack of cartilage hypertrophy and acquisition of the superficial zone proteins such as PRG4/lubricin indicates that engrafted cells, at least in part, have acquired articular chondrocyte identity.

Unfortunately, the described approach where miniature 1 mm defects are generated in articular cartilage in monkeys are not clinically relevant, cannot be considered critical size and do not reproduce any significant change in biomechanics observed in cartilage lesions in humans. The number of animals used in this study is insufficient to assess cartilage restoration statistically and the critical biomechanical assessment of the implanted tissue is missing. The presented 17-week data is insufficient to assess sustainability of the implant, a minimum 6-month time point is required for any therapeutically relevant claims. Thus, presented data represent an elegant in vivo differentiation system while the main translational advantage of the utilized large animal model is missing.

This group has previously presented data where they showed implantation of human pluripotent stem cell-derived aggregates in chondral lesions in pig. Current study offers some technological advancements mentioned as strengths above, but conceptual advantage is incremental and an insufficient effort was made to increase the clinical relevance of the current technology compared to the previous paper.

Post-engraftment analysis of the implants is interesting, but the power of Single-cell sequencing is not fully utilized. It will be useful to see the resulting subsets in comparison with native; analysis of the key chondrogenic genes in native and de-novo iPSC-derived cartilage will be highly relevant.

Presented data on the role of Sik3 in regulating Prg4 expression in mice and cultured cells are interesting, but the mechanistic role of Sik3 in regulating Prg4 in chondrocytes after their transplantation to monkeys is not clear. The authors say that the “results collectively suggest the involvement of SIK3 inactivation in the induction of PRG4 expression in the superficial part of the post-implant cyiPS-Cart”. But completely different mechanism may be co-regulating both Sik3 and Prg4 after

transplantation; if the authors want to claim this mechanism after transplantation, an additional study with genetically modified (Sik3 deletion) chondrocytes is strongly suggested.

The authors did not cite some previously published studies (e.g. Petrigliano et al. 2021) in this field where critical size cartilage lesions were repaired by human PSC-derived chondrocytes with long term engraftment without any immunosuppressive drugs and clinically-relevant structural and functional outcomes. It will be beneficial to specify advantages and limitations of the current study in the context of the current state of the art.

Overall, strength and weaknesses of the study are balanced; presented cell biology data are impressive, but a very substantial amount of work in monkeys will be required to back up any therapeutically claims.

Reviewer #2 (Remarks to the Author):

Abe et al present a paper discussing application of allogeneic induced pluripotent stem (iPS) in chondral defects in a primate animal model. They differentiated GFP-labeled cynomolgus monkey iPS (cyiPSC) cells into chondrocytes and then cultured them for accumulation of extracellular matrix and cartilaginous particles to produce a cyiPSC-derived cartilage organoid (cyiPS-Cart). The authors proceeded by creating chondral defect in the femoral trochlear ridge of the right knee joint of 12 monkeys. cyiPS-Cart were transplanted in 6 monkeys, the other defects remained empty. Animals were sacrificed at 4 and 17 weeks after surgery. Defect sides were processed for histological analysis and scRNA-seq analysis. Accumulation of CD3+ T lymphocytes as a marker for immune reaction was assessed via histology and immunostaining for CD3. The authors were able to show that allogeneic transplantation of cyiPS-Cart did not elicit an immune reaction in chondral defects post-transplantation with allogeneic cyiPS, whereas an immune reaction was observed in osteochondral defects post-transplantation after 4 weeks. They demonstrated that chondral defects in the transplantation group were filled with cartilaginous tissue 4 and 17 weeks after surgery compared to fibrous tissue in the control group. Transplantation of cyiPS-Cart also prevented degeneration of native articular cartilage surrounding the chondral defects. Single-cell RNA sequencing revealed increased expression of pluripotency markers in pre-transplant cyiPSC and chondrocyte markers in pre-transplant cyiPS-Cart. The authors identified upregulation of proteoglycan 4 (PRG4) in post-transplant cyiPS-Cart comparable to levels in control groups without chondral defects. Moreover, increased TGF- β signaling and decreased SIK-signaling was suggested to be involved in PRG4 activation in post-transplant compared to pre-transplant cyiPS-Cart. After application of fluid flow shear stress to chondrocytes, to mimic in vivo articulation in a joint, expression of Prg4 was further increased after 12 hours. Successful engraftment of transplanted cyiPS-Cart was shown by consistent GFP expression and acquisition of PRG4 expression in post-transplant cyiPS-Cart.

I believe this is a strong paper successfully demonstrating engraftment of allogeneic iPS in chondral defects leading to increased cartilage regeneration and tissue repair. However, we ask the author's to comment on a few remarks:

Fig. 1:

o Can the authors elaborate on what tool was used to create the chondral defects in a consistent manner?

o Fig. 1 c shows three chondral defects per animal.

According to the authors, one defect was subjected to histological analysis and the other defect to single-cell RNA sequencing. What happened to the third defect?

- Fig. 7

o In Fig. 7c, the author's show increased PRG4 expression after treatment with TGF β 1. In the text on page 10, they also mention downregulation of PRG4 expression after treatment with TGF- β -inhibitor. What inhibitor was used? Can the author's show the data demonstrating decreased PRG4 expression after treatment with TGF- β -inhibitor?

- Fig. 8

o In Figure 8d, the authors show Prg4 expression in a larger area of the joint in a conditional knockout model lacking Sik3 expression in chondrocytes. The authors later suggest that acquisition of Prg4 expression in post-implant cyiPS-Cart is indicative for successful engraftment. Have the authors performed transplantation of cyiPS-Cart in chondral defects in the knockout mouse model? If the authors hypothesis is correct, engraftment, as indicated by positive GFP staining, should not occur.

- Did the authors perform any functional testing with live animals to demonstrate improvement of function after transplantation of cyiPS-Cart? I recognize this may not be possible, because of the animals having been sacrificed, if the authors have these data, it would be beneficial to include it.

Minor comments:

- Page 5-6, line 117-118: I suggest to use more scientific language and replace the word "stuff"

- Page 8, line 179: I believe you are referring to Figure 2a instead of Figure 3a. Figure 3a does not show immunostaining for CD3 immune cells, but rather shows safranin O, H&E and picrosirius red staining

- Page 8, line 180: As far as I understood, the immune reaction was only assessed after 4 weeks, not 4 months? Please elaborate.

- Page 9, line 216: Do the authors mean:...became articular “cartilage”...instead of articular “cartilaginous”

- Page 12, line 309: Please review that sentence. "facilitate" or "contribute"

Point-by-point response

Reviewer #1 (Remarks to the Author):

In this study, Abe et al. analyzed the allogenic transplantation of MHC-mismatched iPSC cell-derived cartilage organoids in a primate animal model without the use of immunosuppressive drugs. The reviewer would like to acknowledge a significant amount and complexity of the current study. Several strengths include novelty of primate iPSC in cartilage field, large animal model of a cartilage lesion and post-implantation Single-cell sequencing of the engrafted tissue. Presented data clearly demonstrate engraftment of the implanted cells, their chondrogenic identity and lack of significant immune response. In addition, lack of cartilage hypertrophy and acquisition of the superficial zone proteins such as PRG4/lubricin indicates that engrafted cells, at least in part, have acquired articular chondrocyte identity.

1. Unfortunately, the described approach where miniature 1 mm defects are generated in articular cartilage in monkeys are not clinically relevant, cannot be considered critical size and do not reproduce any significant change in biomechanics observed in cartilage lesions in humans. The number of animals used in this study is insufficient to assess cartilage restoration statistically and the critical biomechanical assessment of the implanted tissue is missing. The presented 17-week data is insufficient to assess sustainability of the implant, a minimum 6-month time point is required for any therapeutically relevant claims. Thus, presented data represent an elegant in vivo differentiation system while the main translational advantage of the utilized large animal model is missing.

We appreciate the reviewer's precise description of the strengths and limitations of our study. We agree that while the strength of our study is the use of primate animals with immune systems similar to humans, there are also several limitations. The weight of cynomolgus monkeys is around 3 kg and they are not as large as mini pigs. We also did not perform biomechanical assessment of the transplanted tissue. Furthermore, an observation period of 17 weeks (4 months) is not long enough to conclude the sustainability of the transplant. We have described these limitations in the discussion section on page 13 as follows: "There are several limitations to this study. Firstly, the cynomolgus monkeys used in this study are not large animals, which makes it difficult to

reproduce changes in biomechanics observed in cartilage lesion in humans. Hence, our results will need to be confirmed in critical-sized defects in larger animal models. Secondly, an observation period of 17 weeks (4 months), as employed in this study, is not long enough to demonstrate the sustainability of a transplant. It is, however, still possible to conclude that engrafted cells had obtained a degree of articular chondrocyte identity by 4 months. A longer observation period, such as two years, would further signify the sustainability and turnover of transplants.”

Thanks to the reviewer’s comment, what we aim to clarify through this study is that allogenic cartilage can achieve engraftment in chondral defects without the use of immunosuppressive drugs in primates. The sustainability of cartilage transplants should ideally be tested in large animal models, such as mini pigs. However, bona fide iPS cells have not been available for these large animals, while xenografting human iPS-cell derived cartilage into mini pigs results in severe immune reactions as described below (please see our response to comment No. 5). Since each animal model has specific limitations, we believe that studies should be conducted in both large animals and primates before therapeutically relevant claims are made. Our study has mainly focused on the study of immune reactions to allogeneic cartilage transplantation.

The observation period of one and four months in our study clarified that the integration of cartilage allografts with host native cartilage was not achieved at one month and was achieved, instead, at 4 months after transplantation. The lack of immune reactions for one and four months after transplantation have lead us to the conclusion that immune reactions are not elicited in allogeneic cartilage transplantation in chondral defects. As suggested, it is interesting to test the sustainability and remodeling of cartilage transplants for longer periods. One- or two-year observations may provide interesting results regarding the fate of engrafted cartilage, and we would like to perform such experiments in future studies.

2. This group has previously presented data where they showed implantation of human pluripotent stem cell-derived aggregates in chondral lesions in pig. Current study offers some technological advancements mentioned as strengths above, but conceptual advantage is incremental and an insufficient effort was made to increase the clinical relevance of the current technology compared to the previous paper.

We previously transplanted human iPSC-derived cartilage into cartilage defects of mini pig knee joints and found severe immune reactions against transplants which were

rejected because of xenografting (please see our response to the comment No. 5). Thus, we administered high doses of immunosuppressive drugs to mini pigs, but the immune reaction could be controlled only for one month. We reported the results of transplantation in mini pigs only for one month after transplantation [3]. Hence, our previous study demonstrated that iPSC-derived cartilage survived for one month in a large animal. The advance of that mini-pig experiments is that iPSC cell-derived cartilage can resist mechanical load, the degree of which is compatible to humans, at least for one month.

As clarified in comment #1 from, we consider that the advancements of our study reside in the evidence that the transplantation of allogeneic iPSC-cartilage evaded immune reaction without the use of immunosuppressive drugs, achieved engraftment, integrated with host native cartilage, and, at least in part, have acquired articular cartilage identity by four months. The use of a primate model and primate iPSC cells enabled us to achieve these advancements which we consider are substantial.

3. Post-engraftment analysis of the implants is interesting, but the power of Single-cell sequencing is not fully utilized. It will be useful to see the resulting subsets in comparison with native; analysis of the key chondrogenic genes in native and de-novo iPSC-derived cartilage will be highly relevant.

We performed additional analysis of scRNA-seq data. At first, we compared native articular cartilage, iPSC-derived cartilage and fibrous tissue. We changed the parameter resolution for clustering (Fig. 6 b, c, d). Then, we compared native articular cartilage and iPSC-derived cartilage post-transplant (Extended Data Fig. 6). We found differentially expressed genes (DEGs) in each cluster and performed canonical pathway analysis using ingenuity pathway analysis (IPA, Qiagen) (Fig. 6 f, g). We also analyzed the expression of chondrogenic marker genes which were previously reported using osteoarthritis samples[1]. We have added the results of new analysis in the results section and in Figures 6 b–g and Extended Data Figures 5 and 6 as shown below:

Then, we analyzed cell subpopulations in cyAC, cyFT, pre-transplanted cyiPS-Cart and post-transplant cyiPS-Cart samples. Using the subset function, we reduced the number of cells to 320 in each sample using the subset function and integrated the samples into a single object¹⁵. We also reduced the dimensions, clustered the cells with a parameter resolution of 0.2, and projected them onto a UMAP plot (Fig. 6b). Cell clustering analysis revealed that cyAC, pre-transplant cyiPS-Cart, and post-transplant cyiPS-Cart had similar transcriptional profiles, whereas FT contained cell clusters with

distinct profiles. cyAC, pre-transplant cyiPS-Cart, and post-transplant cyiPS-Cart were composed of cluster #0, whereas FT was composed of clusters #1 and #2 (Fig 6c, d). Cells in clusters #0 and #2 exhibited high expression of *COL2A1* whereas those in cluster #1 highly expressed *COL1A1* (Fig. 6e). Differentially expressed genes (DEGs) were identified (Fig. 6f). Canonical pathway analysis based on the DEGs indicated that cluster #1 was enriched for the fibrosis pathway and that cluster #2 was enriched for the osteoarthritis pathway (Fig. 6g). These results suggest that cluster #1 and #2 consisted of pathological cells and contained few cyAC, pre-transplant cyiPS-Cart, and post-transplant cyiPS-Cart cells (Fig. 6d).

Figure 6

Figure 6. scRNA-seq analysis of cyAC, cyFT, pre-transplant cyiPS-Cart, and post-transplant cyiPS-Cart.

- a) The VlnPlot (Seurat) shows the distribution of single-cell gene expression in each sample. The y-axis of each panel represents the expression levels of the indicated genes.
- b) After reducing the cell number for each sample to 320, the data from the samples were integrated. The cells were then clustered with a parameter resolution of 0.2 and

projected onto the UMAP plots.

- c) UMAP plot in (b) separated by samples.
- d) The ratio of the number of cells in each cell cluster in each sample (c) is plotted.
- e) *COL2A1* and *COL1A1* expression levels are indicated in each cell projected on the UMAP plot using the feature plot function.
- f) Heatmap revealing the scaled expression of differentially expressed genes for each cluster defined in (b).
- g) Canonical pathways enriched for each cluster based on differentially expressed genes. The results of Clusters #3 and #4 were omitted because there were very few cells in these clusters.

We further compared these cell clusters with those previously identified in human osteoarthritis samples [1]. Expression of marker genes for osteoarthritis [1] in our clusters suggested that cluster #1 corresponds to preHTC and FC (high expression of *TGFBI* and *COL1A1*) and that cluster #2 corresponds to EC and proC (high expression of *TF* and *P3H2*; Extended Data Fig. 5 a, b).

Extended Data Figure 5

Extended Data Figure 5. scRNA-seq analysis of cyAC, cyFT, pre-transplant cyiPS-Cart, and post-transplant cyiPS-Cart.

- a) Expression levels of *COL2A1* and osteoarthritis marker genes are indicated in each cell projected on the UMAP plot using the feature plot function.
- b) Expression levels of *COL2A1* and osteoarthritis marker genes are indicated in each cell projected on the UMAP plot using the “VlnPlot” function. The results of Clusters #3 and #4 were omitted because cell numbers in these clusters are few.

To analyze the resemblance and difference between cyAC and post-transplant cyiPS-Cart cells, we selected these cells and performed clustering analysis again. Cells were divided into four clusters (Extended Data Fig. 6 a, b). Clusters #0, 1 and 2 consisted of both cyAC

and post-transplant cyiPS-Cart cells, whereas post-transplant cyiPS-Cart cells were excluded from cluster #3 (Extended Data Fig. 6 a, c, and d). DEGs and canonical pathway analysis indicated that cluster #3 was enriched for integrin signaling (Extended Data Fig. 6 e, f). Trajectory inference and RNA velocity analysis suggested that cluster #3 was located at the start of the trajectory to #0 (Extended Data Fig. 6g), whereas cluster #2 was located at the end. These results suggest that post-transplant cyiPS-Cart are similar to cyAC except for cells from cluster #3 that are related to integrin signaling and located at the start of the trajectory.

Extended Data Figure 6

Extended Data Figure 6. scRNA-seq analysis of cyAC, cyF and post-transplant cyiPS-Cart.

a) After reducing the cell number for each sample to 320, the data from the samples were integrated. The cells were then clustered with a parameter resolution of 0.95 and

projected onto the UMAP plots.

- b) *COL2A1* and *COL1A2* expression levels are indicated in each cell projected on the UMAP plot using the feature plot function.
- c) Distribution of cells in each sample is indicated on the UMAP plot.
- d) The ratio of the number of cells in each cell cluster in each sample in (a) is plotted.
- e) Heatmap revealing the scaled expression of differentially expressed genes for each cluster defined in (a).
- f) Canonical pathways enriched for each cluster based on differentially expressed genes.
- g) Post-transplant cyiPS-Cart cells were subjected to RNA velocity analysis using scVelo. The trajectory inference using PAGA [5] was extended by the velocity-inferred directionality.

4. Presented data on the role of *Sik3* in regulating *Prg4* expression in mice and cultured cells are interesting, but the mechanistic role of *Sik3* in regulating *Prg4* in chondrocytes after their transplantation to monkeys is not clear. The authors say that the “results collectively suggest the involvement of *SIK3* inactivation in the induction of *PRG4* expression in the superficial part of the post-transplant cyiPS-Cart”. But completely different mechanism may be co-regulating both *Sik3* and *Prg4* after transplantation; if the authors want to claim this mechanism after transplantation, an additional study with genetically modified (*Sik3* deletion) chondrocytes is strongly suggested.

Thank you for appreciating our study in relationship between *Sik3* and *Prg4* expression. We demonstrated that *Sik3* deletion in chondrocytes increased *Prg4* expression cartilage in mice in knockout and transgenic mice (Fig. 8 a, c, d; Extended Data Fig. 7a) and that *Sik3* deletion in chondrocytes under shear stress increased *Prg4* expression *in vitro* (Fig. 8e; Extended Data Fig. 7 b, c). These results indicate that *Sik3* inhibits *Prg4* expression in mice, but does not guarantee that *SIK3* inhibits *PRG4* expression in the post-transplant cyiPS-Cart in monkey. To prove this, we may need to perform additional experiments of transplanting cyiPS-Cart in which the *SIK3* gene is deleted into chondral defects of cynomolgus monkey knee joints and confirming increased *PRG4* expression in the post-transplant *SIK3* knockout cyiPS-Cart. However, it is difficult to perform such experiments in monkeys because they are highly technically demanding. It is also difficult to obtain cynomolgus monkeys for experiments because of tight international supply, partly due to COVID-19 restrictions. Thus, we deleted the sentence “**These results collectively suggest the involvement of *SIK3* inactivation in the**

induction of PRG4 expression in the superficial part of the post-implant cyiPS-Cart.” from the results section (page 10). Instead, we added following sentence as the limitation of our study in the discussion section (page 13): “...although our results suggest that Sik3 inhibits *Prg4* expression in mouse chondrocytes, this does not prove that SIK3 is involved in *PRG4* expression in post-transplant cyiPS-Cart in monkeys.”

In addition, we corrected sentences as below:

“Pathway analysis suggested the involvement of SIK3 inactivation, verified through molecular experiments.” has been changed to “Pathway analysis suggested the involvement of SIK3 inactivation.” in the abstract (page 2)

“Sik3 plays an important role in regulating *Prg4* expression in vivo.” has been changed to “Sik3 helps regulate *Prg4* expression in mouse chondrocytes.” in the results section (page 10).

“SIK3 is involved in post-transplant differentiation” has been changed to “SIK3 could be involved in post-transplant differentiation” in the discussion section (page 12).

To obtain insight into the upstream regulators of Sik3 activities and *Prg4* expression, we analyzed the relationship between TGF- β and Sik3 because TGF- β increased *PRG4* expression in cyiPS-Cart cells (Fig. 7e). The addition of TGF- β did not affect phosphorylation of Sik3 at T411. The addition of forskolin increased phosphorylation of Sik3 at T411 but did not affect phosphorylation of Smad3. These results suggest that TGF- β and Sik3 regulate *Prg4* expression independently. We have added these results in Extended Data Fig. 8 and the results section (page 11).

Extended Data Figure 8. Relationship between TGF- β signaling and Sik3 in chondrocytes.

Mouse primary chondrocytes were treated with or without 1, 5 or 10 ng/ml TGF β 1 (left), or treated with or without 1, 5 or 10 ng/ml TGF β 1 or 100 μ g/ml forskolin (right) for 30 min for immunoblot analysis.

5. The authors did not cite some previously published studies (e.g. Petrigliano et al. 2021) in this field where critical size cartilage lesions were repaired by human PSC-derived chondrocytes with long term engraftment without any immunosuppressive drugs and clinically-relevant structural and functional outcomes. It will be beneficial to specify

advantages and limitations of the current study in the context of the current state of the art.

Thank you for referring us to this important study. As we have described in the response to comment no. 2 above, the transplantation of human iPS-derived cartilage into articular cartilage defects of mini pig caused severe immune reactions, and the transplants were rejected. Only small part of transplants later remained (please see the Figure below, unpublished data).

Fig. Human iPS cell-derived cartilage (hiPS-Cart) was transplanted into articular cartilage defect in the knee joint of a mini-pig. Xenograft of hiPS-Cart caused severe immune reaction and was rejected, leaving a trace of hiPS-Cart transplant. Lymphocytes accumulated around the hiPS-Cart transplant. Bottom, semi-serial section of top right panel was subjected to immunohistochemistry using antibody that recognize only human vimentin but not pig vimentin. A trace of hiPS-Cart transplant was detected.

Similarly, Petrigliano et al. [2] showed that only 4% of total cells in the defects are human cells (Fig. 4a,b, Petrigliano et al. 2021 [2]), indicating that 96% of cells in the repair tissue in the defects were mini pig cells. This mean that most human ES cell-derived chondrocytes were lost after implantation and that host mini pig cells constituted the repair tissue in the articular cartilage defects. We speculate from those results that the implanted ESC-derived chondrocytes could survive only shortly. The implanted human

ESC-derived chondrocytes might secrete factors that stimulate mini-pig progenitor cells that subsequently form repair tissue in the defects (trophic effects).

We believe that the advantage of our study is that almost all cells in repair tissue are transplanted cells, suggesting that transplants directly contribute to the repair tissue. On the other hand, the limitations of our study include that our results are from cynomolgus monkeys that are relatively small compared to mini pigs. This repair mechanism of direct contribution should be tested in large animal models. We have added this discussion in the discussion section as follows: “One recent study implanted human embryonic stem cell (ESC)-derived chondrocytes that do not associate with cartilage ECM into the articular cartilage defects of mini-pigs³². Of the cells formed in repaired tissue in the defects, 4% were human, indicating that most cells that form in repair tissue are those of the transplant recipient. In contrast, in our study, allogeneic cyiPS-Cart survived for at least four months, and almost all cells in the repair tissue were transplanted cells as demonstrated by GFP expression.” (page 12) and “There are several limitations to this study. Firstly, the cynomolgus monkeys used in this study are not large animals, which makes it difficult to reproduce changes in biomechanics observed in cartilage lesion in humans. Hence, our results will need to be confirmed in critical-sized defects in larger animal models. Secondly, an observation period of 17 weeks (4 months), as employed in this study, is not long enough to demonstrate the sustainability of a transplant. It is, however, still possible to conclude that engrafted cells had obtained a degree of articular chondrocyte identity by 4 months.” (page 13).

6. Overall, strength and weaknesses of the study are balanced; presented cell biology data are impressive, but a very substantial amount of work in monkeys will be required to back up any therapeutically claims.

Thank you for your detailed assessment of our study. In order to strengthen the work in monkeys as much as possible, we performed additional experiments in which the presence or absence of lymphocytes was analyzed around transplants four months after transplantation by immunohistochemistry (Fig. 4b). Lymphocyte accumulations were not detected, confirming that immune reactions did not occur at four months after transplantation of allogeneic cartilage in chondral defects. In addition, to further support cell biology data, we added experimental results from treatment of cyiPS-derived cartilage chondrocytes with TGF- β inhibitor SB431542 (Fig. 7e, right). The inhibition of TGF- β signals decreased PRG4 expression, supporting the result that the addition of TGF- β 1 to the culture of cyiPS-Cart cells increased *PRG4* mRNA expression.

Figure 4b. Semi-serial histological sections of samples at 17 weeks after transplantation were immunostained for CD3.

Figure 7e, right. Cells from the pre-transplant cyiPS-Cart were cultured in the presence or absence of TGF-β1 (*left*) or TGF-β inhibitor, SB431542 (*right*). *PRG4* mRNA expression was analyzed using real-time RT-PCR. Error bars denote means ± SE. ** $P = 0.0048$, ** $P = 0.0017$ by two-tailed Student's *t*-test ($n = 3$).

Reviewer #2 (Remarks to the Author):

Abe et al present a paper discussing application of allogeneic induced pluripotent stem (iPS) in chondral defects in a primate animal model. They differentiated GFP-labeled cynomolgus monkey iPS (cyiPSC) cells into chondrocytes and then cultured them for accumulation of extracellular matrix and cartilaginous particles to produce a cyiPSC-

derived cartilage organoid (cyiPS-Cart). The authors proceeded by creating chondral defect in the femoral trochlear ridge of the right knee joint of 12 monkeys. cyiPS-Cart were transplanted in 6 monkeys, the other defects remained empty. Animals were sacrificed at 4 and 17 weeks after surgery. Defect sides were processed for histological analysis and scRNA-seq analysis. Accumulation of CD3+ T lymphocytes as a marker for immune reaction was assessed via histology and immunostaining for CD3. The authors were able to show that allogeneic transplantation of cyiPS-Cart did not elicit an immune reaction in chondral defects post-transplantation with allogeneic cyiPS, whereas an immune reaction was observed in osteochondral defects post-transplantation after 4 weeks. They demonstrated that chondral defects in the transplantation group were filled with cartilaginous tissue 4 and 17 weeks after surgery compared to fibrous tissue in the control group. Transplantation of cyiPS-Cart also prevented degeneration of native articular cartilage surrounding the chondral defects. Single-cell RNA sequencing revealed increased expression of pluripotency markers in pre-transplant cyiPSC and chondrocyte markers in pre-transplant cyiPS-Cart. The authors identified upregulation of proteoglycan 4 (PRG4) in post-transplant cyiPS-Cart comparable to levels in control groups without chondral defects. Moreover, increased TGF- β signaling and decreased SIK-signaling was suggested to be involved in PRG4 activation in post-transplant compared to pre-transplant cyiPS-Cart. After application of fluid flow shear stress to chondrocytes, to mimic in vivo articulation in a joint, expression of Prg4 was further increased after 12 hours. Successful engraftment of transplanted cyiPS-Cart was shown by consistent GFP expression and acquisition of PRG4 expression in post-transplant cyiPS-Cart.

I believe this is a strong paper successfully demonstrating engraftment of allogeneic iPS in chondral defects leading to increased cartilage regeneration and tissue repair. However, we ask the author's to comment on a few remarks:

Fig. 1:

1. Can the authors elaborate on what tool was used to create the chondral defects in a consistent manner?

We used the electric router, Dremel micro (please see figure below), and a dental steel bur under a surgical loupe to create chondral defects in a consistent manner. We have added this information in the methods section (page 16).

Figure. Dremel, micro and dental steel bar.

1. Fig. 1 c shows three chondral defects per animal.

According to the authors, one defect was subjected to histological analysis and the other defect to single-cell RNA sequencing. What happened to the third defect?

We apologize for the confusion. One defect was subjected to histological analysis. To secure the number of cells for scRNA-seq analysis, we harvested two defects and combined them. We have now rewritten the results (page 6) and methods sections (page 16) to remove the discrepancy.

- Fig. 7

3. In Fig. 7c, the author's show increased PRG4 expression after treatment with TGFB1. In the text on page 10, they also mention downregulation of PRG4 expression after treatment with TGF- β -inhibitor. What inhibitor was used? Can the author's show the data demonstrating decreased PRG4 expression after treatment with TGF- β -inhibitor?

We apologize for the confusion. We have added experimental results from the treatment of cyiPS-Cart cells with TGF- β inhibitor (SB 431542). The inhibition of TGF- β signals decreased PRG4 expression, supporting the result that the addition of TGF- β 1 to the culture of cyiPS-Cart cells increased *PRG4* mRNA expression. We have added the data to Fig. 7e and rewritten the results section (page 10).

Figure 7e, right. Cells from the pre-transplant cyiPS-Cart were cultured in the presence or absence of TGF-β1 (*left*) or TGF-βinhibitor SB431542 (*right*). *PRG4* mRNA expression was analyzed using real-time RT-PCR. Error bars denote means ± SE. ***P* = 0048, ***P* = 0.0017 by two-tailed Student’s *t*-test (n = 3).

In addition, we have corrected the *P* value in the results of experiments in which TGF-β1 was added to cyiPS-Cart cells in Fig 7e.

- Fig. 8

4. In Figure 8d, the authors show *Prg4* expression in a larger area of the joint in a conditional knockout model lacking *Sik3* expression in chondrocytes. The authors later suggest that acquisition of *Prg4* expression in post-implant cyiPS-Cart is indicative for successful engraftment. Have the authors performed transplantation of cyiPS-Cart in chondral defects in the knockout mouse model? If the authors hypothesis is correct, engraftment, as indicated by positive GFP staining, should not occur.

We are unsure whether we could correctly follow your comment. We would like to point out that we have not transplanted cyiPS-Cart into articular cartilage defects created in the knee joints of *Sik3* conditional knockout mice. We have observed that the phrase “Acquisition of PRG4 expression in post-transplant cyiPS-Cart” in the results section (page 11) is misleading. Thus, we have corrected this phrase to “Restricted expression of PRG4 in superficial zone in post-transplant cyiPS-Cart”.

5. - Did the authors perform any functional testing with live animals to demonstrate improvement of function after transplantation of cyiPS-Cart? I recognize this may not be

possible, because of the animals having been sacrificed, if the authors have these data, it would be beneficial to include it.

Thank you for your valuable input. Unfortunately, we did not perform limb function testing with live animals. When we designed the experiments, we attempted to quantitatively assess the movement of monkeys to assess the function of their knee joints. We considered using motion recorders and mats to sense load from each of the monkey's limbs. However, such an assessment of motor function was not successful, and we could not employ it in the study. We wish to incorporate such an assessment in future studies.

Minor comments:

- Page 5-6, line 117-118: I suggest to use more scientific language and replace the word “stuff”

We have changed the word to “tissue”.

- Page 8, line 179: I believe you are referring to Figure 2a instead of Figure 3a. Figure 3a does not show immunostaining for CD3 immune cells, but rather shows saffranin O, H&E and picosirius red staining

- Page 8, line 180: As far as I understood, the immune reaction was only assessed after 4 weeks, not 4 months? Please elaborate.

We additionally performed immunostaining for CD3 in the samples obtained 4 months after transplantation. We have added the data in Fig. 4b and in the results section (pages 7 and 8).

Figure 4b. Semi-serial histological sections of samples at 17 weeks after transplantation were immunostained for CD3.

In addition, we have changed an image in the bottom row and middle column in Figure 2a.

- Page 9, line 216: Do the authors mean:...became articular “cartilage”...instead of articular “cartilaginous”

We rewrote the phrase to “cyiPS-Cart became similar to articular cartilage after transplantation,” because we did not prove that cyiPS-Cart became identical to articular cartilage.

- Page 12, line 309: Please review that sentence. "facilitate" or "contribute"

Thank you. We have deleted the word “facilitate”.

References

1. Ji, Q., Zheng, Y., Zhang, G., Hu, Y., Fan, X., Hou, Y., Wen, L., Li, L., Xu, Y., Wang, Y., and Tang, F., Single-cell RNA-seq analysis reveals the progression of human osteoarthritis. *Ann Rheum Dis*, 78(1): p. 100-110, 2019.
2. Petrigliano, F.A., Liu, N.Q., Lee, S., Tassej, J., Sarkar, A., Lin, Y., Li, L., Yu, Y., Geng, D., Zhang, J., Shkhyan, R., Bogdanov, J., Van Handel, B., Ferguson, G.B., Lee, Y., Hinderer, S., Tseng, K.C., Kavanaugh, A., Crump, J.G., Pyle, A.D., Schenke-Layland, K., Billi, F., Wang, L., Lieberman, J., Hurtig, M., and Evseenko, D., Long-term repair of porcine articular cartilage using cryopreservable, clinically compatible human embryonic stem cell-derived chondrocytes. *NPJ Regen Med*, 6(1): p. 77, 2021.
3. Yamashita, A., Morioka, M., Yahara, Y., Okada, M., Kobayashi, T., Kuriyama, S., Matsuda, S., and Tsumaki, N., Generation of Scaffoldless Hyaline Cartilaginous Tissue from Human iPSCs. *Stem Cell Reports*, 4(3): p. 404-418, 2015.
4. Stuart, T., Butler, A., Hoffman, P., Hafemeister, C., Papalexi, E., Mauck, W.M., III, Hao, Y., Stoeckius, M., Smibert, P., and Satija, R., Comprehensive Integration of Single-Cell Data. *Cell*, 177(7): p. 1888-1902.e1821, 2019.
5. Wolf, F.A., Hamey, F.K., Plass, M., Solana, J., Dahlin, J.S., Göttgens, B., Rajewsky, N., Simon, L., and Theis, F.J., PAGA: graph abstraction reconciles clustering with trajectory

inference through a topology preserving map of single cells. *Genome Biol*, 20(1): p. 59, 2019.

REVIEWER COMMENTS

Reviewer #1 (Remarks to the Author):

The revised manuscript by Abe et al. describes engraftment of allogeneic iPS cell-derived cartilage organoid in a primate model of articular cartilage defect. As initially mentioned by the reviewer, this manuscript represents some important advancements in the area of stem cell-based cartilage repair and may, in a mid-long term, result in clinically-relevant therapeutic intervention.

The main strength of this study, from this reviewer's opinion, is the post-transplantation analysis of the implanted cell identity using single cell seq-based techniques. This type of studies has not been done in the area of cartilage transplantation. This is very valuable for our understanding of the transplanted cell fate in vivo; it also highlights the importance of the functional niche where multiple biochemical and biomechanical signals can guide the final specification of stem cell derived implant.

Additional comments.

The authors adequately discussed some of the limitations of the study, such as short-term timeline, but some additional corrections are necessary. The reviewer understands and accepts the fact that it was not practically possible to conduct additional biomechanical analysis in these animals. However, biomechanical component is critical of the assessment of clinical relevance. . Lack of biomechanical analysis needs of cell related vs control defect need to be mentioned as a limitation. The authors should also mention that 1 mm defects are not clinically relevant as they are not critical size defects, and the results presented in Figure 3 and 4 clearly indicate that non-treaded defects can spontaneously heal to a satisfactory level, especially by 4 months. The defects will probably be completely healed in a 6-12-month interval not presented in the paper which questions any advantages of cell therapy in this model, and is a major issue. Despite the reviewer's enthusiasm about the scientific and technological parts of the study, the clinical relevance needs strengthening and should be clearly presented to the readers.

The implants were made in trochlea, which is not a typical load-bearing region of the joint. "Chondral defects (1 mm diameter and 0.5 mm depth) were created at the trochlea of the distal femur under surgical microscopy". The schematics in Figure 1 do not reflect that, and the way it appears in the image is misleading. It seems that the defect is in the middle of a condyle, which is not appropriate, needs to be corrected in the schematics and stated more clearly.

Defects presented in Figure 2 and 3 make an impression that the Empty defect is much deeper; the Empty defect looks like a full-size chondral defect while the cell-treated looks like a partial thickness

defect (the boundary of the implant is very clear). It is suggested to select a better matching pictures or to mention that the implant induces some response from endogenous cartilage under the implant.

The authors highlighted in the discussion that percentage of the engrafted cells in their studies is higher than in previous publications, which is appropriate. But this statement is completely taken out of the context; many other parameters of these studies are not the same – in the cited study (#32) by Petrigliano et al., the group utilized critical size large defects, the model is xenograft-based, and analysis has been conducted at 6 months post transplantation. The authors do not present 6-month data and it is not clear how would these implants behave in a critical size defect. This needs to be addressed in the text. Otherwise, this discussion is very helpful and shows some advantages of the presented work.

The authors should be also clearer about the functional importance of the percentage of engrafted cells. Any point of view will be acceptable by this reviewer, but it needs to be presented discussed. Do the Authors believe that this is therapeutically important? As mentioned above, cellular composition of the defect may be much less important than the structural and biomechanical restoration of the defects. If the authors disagree and insist that percentage of ingrafted cells directly correlates with therapeutically relevant outcomes, this needs to be stated and justified by some experimental evidence such as a direct correlation between the % of engrafted cells and functional quality of the tissue (e.g. biomechanics), as well as other functional outcomes (e.g. pain). This is a central part of the study as the final destination of this research is human articular cartilage repair.

The reviewer agrees with the strategy regarding the role of SIK3; it is a suggestive mechanism and the authors have now adjusted the claims accordingly.

Overall, the reviewer feels that the manuscript now only requires textual, but very important revisions outlined above, and after those are completed, it can be considered for publication

Reviewer #2 (Remarks to the Author):

We thank the authors for their revisions. However, there are two major points that remain insufficiently addressed.

In Figure 8, they show that *Sik3* deletion leads to increased *Prg4* expression in chondrocytes from knockout mice. However, this was not confirmed in an additional in vivo study, which is why the authors

rephrased their claim of involvement of SIK3 inactivation in the induction of PRG4 in the post-implant cyiPS-Cart.

In addition, biomechanical assessment remains missing. The authors explain that attempts to assess function of the knee joints using motion recorders and mats to sense load has been unsuccessful.

Additionally, the authors state difficulties in acquisition of more monkeys due to tight international supply as well as high technical demand as reasons for not performing the above named experiments.

We understand these limitations and appreciate the author's acknowledgement in the discussion section.

Even though we acknowledge the difficulties the authors face in performing additional work in primates, we do not believe that the current data presented meets the standards for publication in this journal.

Point-by-point response

We thank the reviewers for their constructive comments that really helped us improve our manuscript. We have responded to each comment below and revised our manuscript accordingly.

Reviewer #1 (Remarks to the Author):

The revised manuscript by Abe et al. describes engraftment of allogeneic iPS cell-derived cartilage organoid in a primate model of articular cartilage defect. As initially mentioned by the reviewer, this manuscript represents some important advancements in the area of stem cell-based cartilage repair and may, in a mid-long term, result in clinically-relevant therapeutic intervention.

The main strength of this study, from this reviewer's opinion, is the post-transplantation analysis of the implanted cell identity using single cell seq-based techniques. This type of studies has not been done in the area of cartilage transplantation. This is very valuable for our understanding of the transplanted cell fate in vivo; it also highlights the importance of the functional niche where multiple biochemical and biomechanical signals can guide the final specification of stem cell derived implant.

Additional comments.

The authors adequately discussed some of the limitations of the study, such as short-term timeline, but some additional corrections are necessary. The reviewer understands and accepts the fact that it was not practically possible to conduct additional biomechanical analysis in these animals. However, biomechanical component is critical of the assessment of clinically relevance. Lack of biomechanical analysis needs of cell related vs control defect need to be mentioned as a limitation. The authors should also mention that 1 mm defects are not clinically relevant as they are not critical size defects, and the results presented in Figure 3 and 4 clearly indicate that non-treated defects can spontaneously heal to a satisfactory level, especially by 4 months. The defects will probably be completely healed in a 6-12-month interval not presented in the paper which questions any advantages of cell therapy in this model, and is a major issue. Despite the reviewer's enthusiasm about the scientific and technological parts of the study, the clinical relevance needs strengthening and should be clearly presented to the readers.

<Response>

We thank reviewers for raising important points that were required to describe our study appropriately.

We added the following description in Discussion section as a limitation of our study: “... Although chondral defects in the empty group were filled with fibrous tissue in a four-month interval, the small size of the defects would eventually heal after a longer period. Hence, our results will need to be confirmed in larger animal models with critical-sized defects. In addition, we did not analyze the biomechanical properties of post-transplant cyiPS-Cart. Biomechanical test will be required to determine clinical relevance.”

The implants were made in trochlea, which is not a typical load-bearing region of the joint. “Chondral defects (1 mm diameter and 0.5 mm depth) were created at the trochlea of the distal femur under surgical microscopy”. The schematics in Figure 1 do not reflect that, and the way it appears in the image is misleading. It seems that the defect is in the middle of a condyle, which is not appropriate, needs to be corrected in the schematics and stated more clearly.

<Response>

The scheme was inappropriate and misleading. We corrected the schematics in Figure 1b.

Defects presented in Figure 2 and 3 make an impression that the Empty defect is much deeper; the Empty defect looks like a full-size chondral defect while the cell-treated looks like a partial thickness defect (the boundary of the implant is very clear). It is suggested to select a better matching pictures or to mention that the implant induces some response from endogenous cartilage under the implant.

<Response>

We thank the reviewer for indicating an important point. We consider that the transplant induces some response from endogenous cartilage under the transplant. We added magnification images of histological sections as Figure 3c. Higher magnification of the histological sections in the empty group indicates that the defects were within cartilage and did not reach into bone. Safranin O staining was lost in the cartilage matrix that locates between the defect and bone in the empty group. We modified description in Results section as follows: “The articular cartilage adjacent to the chondral defect in the empty group lost safranin O staining at 17 weeks after surgery (Fig. 3a, *arrows*; Supplementary Fig. 4b). The remaining cartilage that locates between the bottom of the

defect and bone also lost safranin O staining at 4 and 17 weeks after surgery in the empty group (Fig. 3c, area below the *dotted lines*). These results indicate progressive degeneration of the articular cartilage around the defects. In contrast, the articular cartilage surrounding the chondral defect maintained proteoglycan in the transplantation group at 17 weeks after surgery, suggesting preservation of articular cartilage around the defects (Fig. 3a, *arrow heads*; Fig. 3c, area below the *dotted lines*; Supplementary Fig. 4b).”

Figure 3c. Magnifications of remaining cartilage located between the bottom of the defect and bone in (a). *Dotted lines* indicate bottom of defects. Safranin O staining. Scale bars, 100 μm .

The authors highlighted in the discussion that percentage of the engrafted cells in their studies is higher than in previous publications, which is appropriate. But this statement is completely taken out of the context; many other parameters of these studies are not the same – in the cited study (#32) by Petrigliano et al., the group utilized critical size large defects, the model is xenograft-based, and analysis has been conducted at 6 months post transplantation. The authors do not present 6-month data and it is not clear how would these implants behave in a critical size defect. This needs to be addressed in the text. Otherwise, this discussion is very helpful and shows some advantages of the presented work.

<Response>

We added the following sentence in Discussion section: “Although experimental conditions between the two studies differ (xenograft vs. allograft; critical vs. small size defects; 6 vs. 4 months observation)”.

The authors should be also clearer about the functional importance of the percentage of engrafted cells. Any point of view will be acceptable by this reviewer, but it needs to be presented discussed. Do the Authors believe that this is therapeutically important? As

mentioned above, cellular composition of the defect may be much less important than the structural and biomechanical restoration of the defects. If the authors disagree and insist that percentage of ingrafted cells directly correlates with therapeutically relevant outcomes, this needs to be stated and justified by some experimental evidence such as a direct correlation between the % of engrafted cells and functional quality of the tissue (e.g. biomechanics), as well as other functional outcomes (e.g. pain). This is a central part of the study as the final destination of this research is human articular cartilage repair.

<Response>

We agree that the importance of the percentage of engrafted cells that survive is not clear. Accordingly, we added the following discussion: “Regarding clinical relevance, it has not been known whether engraftment of cartilage transplants gives better clinical results, such as improved joint function and pain relief, than repair tissue formed by trophic effects or vice versa. It is plausible that engraftment of cartilage transplants is better indicated for severe cartilage lesions where the provision of host progenitor cells is limited. Further study is required to clarify indications.”

The reviewer agrees with the strategy regarding the role of SIK3; it is a suggestive mechanism and the authors have now adjusted the claims accordingly.

Overall, the reviewer feels that the manuscript now only requires textual, but very important revisions outlined above, and after those are completed, it can be considered for publication

Reviewer #2 (Remarks to the Author):

We thank the authors for their revisions. However, there are two major points that remain insufficiently addressed.

In Figure 8, they show that *Sik3* deletion leads to increased *Prg4* expression in chondrocytes from knockout mice. However, this was not confirmed in an additional in vivo study, which is why the authors rephrased their claim of involvement of SIK3 inactivation in the induction of PRG4 in the post-implant cyiPS-Cart.

In addition, biomechanical assessment remains missing. The authors explain that attempts to assess function of the knee joints using motion recorders and mats to sense load has been unsuccessful.

Additionally, the authors state difficulties in acquisition of more monkeys due to tight international supply as well as high technical demand as reasons for not performing

the above named experiments.

We understand these limitations and appreciate the author's acknowledgement in the discussion section.

Even though we acknowledge the difficulties the authors face in performing additional work in primates, we do not believe that the current data presented meets the standards for publication in this journal.

We appreciate your valuable feedback. In the revised manuscript, we have tried our best to address all your concerns and improve the manuscript as per your suggestions and feedback. We hope that you find the revised manuscript suitable for publication.

REVIEWERS' COMMENTS

Reviewer #1 (Remarks to the Author):

All issues in the text and images are addressed.